# Skin Wound Healing of the Adult Newt, *Cynops pyrrhogaster*: A Unique Re-Epithelialization and Scarless Model

**DOI:** 10.3390/biomedicines9121892

**Published:** 2021-12-13

**Authors:** Tatsuyuki Ishii, Ikkei Takashimizu, Martin Miguel Casco-Robles, Yuji Taya, Shunsuke Yuzuriha, Fubito Toyama, Fumiaki Maruo, Kazuo Kishi, Chikafumi Chiba

**Affiliations:** 1Department of Plastic and Reconstructive Surgery, Keio University, Shinanomachi 35, Tokyo 160-8582, Japan; ttsyksh@keio.jp; 2Department of Plastic and Reconstructive Surgery, Shinshu University School of Medicine, Asahi 3-1-1, Matsumoto 390-8621, Japan; ikkeit@shinshu-u.ac.jp (I.T.); yuzuriha@shinshu-u.ac.jp (S.Y.); 3Faculty of Life and Environmental Sciences, University of Tsukuba, Tennodai 1-1-1, Tsukuba 305-8572, Japan; casco.miguel.gm@u.tsukuba.ac.jp (M.M.C.-R.); maru@biol.tsukuba.ac.jp (F.M.); 4Department of Pathology, The Nippon Dental University School of Life Dentistry at Tokyo, Fujimi 1-9-20, Tokyo 102-8159, Japan; taya-yu@tky.ndu.ac.jp; 5Graduate School of Engineering, Utsunomiya University, Yoto 7-1-2, Utsunomiya 321-8585, Japan; fubito@is.utsunomiya-u.ac.jp

**Keywords:** newt, skin regeneration, re-epithelialization, scarless, color pattern

## Abstract

In surgical and cosmetic studies, scarless regeneration is an ideal method to heal skin wounds. To study the technologies that enable scarless skin wound healing in medicine, animal models are useful. However, four-limbed vertebrates, including humans, generally lose their competency of scarless regeneration as they transit to their terrestrial life-stages through metamorphosis, hatching or birth. Therefore, animals that serve as a model for postnatal humans must be an exception to this rule, such as the newt. Here, we evaluated the adult newt in detail for the first time. Using a Japanese fire-bellied newt, *Cynops pyrrhogaster*, we excised the full-thickness skin at various locations on the body, and surveyed their re-epithelialization, granulation or dermal fibrosis, and recovery of texture and appendages as well as color (hue, tone and pattern) for more than two years. We found that the skin of adult newts eventually regenerated exceptionally well through unique processes of re-epithelialization and the absence of fibrotic scar formation, except for the dorsal-lateral to ventral skin whose unique color patterns never recovered. Color pattern is species-specific. Consequently, the adult *C. pyrrhogaster* provides an ideal model system for studies aimed at perfect skin wound healing and regeneration in postnatal humans.

## 1. Introduction

After the skin of adult mammals is wounded, it heals but never regenerates completely. The last stage of wound healing is scar formation. Apart from normal skin, scars can annoy patients cosmetically and sometimes functionally. Thus, skin regeneration without leaving scars after traumas or surgeries is one of the most challenging issues that surgeons are pursuing [1].

Skin, the largest organ that envelops the body, is essential for the regulation of body temperature and water, offers protection against invasion of pathogens, provides sensation to environmental cues, and sends signals for social communication [2]. Therefore, wounding of the skin can sometimes impact the quality of life. In general, skin is endowed with the ability to heal itself after a traumatic injury by closing the wound via the action of the epidermis, which grows from the margins of the wound in a process referred to as re-epithelialization. However, depending on the severity of damage, the texture (pattern of sulcus cutis and cristae cutis), appendages (e.g., hairs and sweat glands) and color (pattern and tone or hue) of skin do not always recover [3]. In reconstructive surgery and cosmetic studies, these test items during skin regeneration must be evaluated.

As a rule, four-limbed vertebrates (tetrapods), including humans, have a relatively high capacity to repair or regenerate their injured body parts in their aquatic life-stages (as larvae (amphibians), embryos (reptiles and birds) or fetuses (mammals)), but this ability declines or is lost as they transit to terrestrial life-stages through metamorphosis (amphibians), hatching (reptiles and birds) or birth (mammals) [4,5,6,7,8,9,10,11,12]. It is notable that in the terrestrial life-stage, missing parts of the body after trauma are healed by creating fibrotic tissues or scars, rather than regenerates [4,6,7,8,10]. This rule is true for skin. In mice, the skin of fetuses can regenerate after wounding until embryonic day (E) 13, but as they grow beyond this stage, they no longer regenerate skin texture [6]. Importantly, fibrosis or fibrotic scar formation contributes to the pathogenesis of various diseases such as pulmonary fibrosis and kidney fibrosis [13]. In skin, granulation followed by dermal fibrosis is known to impact the recovery of its texture, appendages and color, eventually resulting in a scar [2]. Therefore, controlling granulation and dermal fibrosis is an important subject in clinical studies of skin regeneration.

To study skin regeneration, animal models are useful. It is generally believed that amphibians have the capacity to effectively regenerate their skin without forming scars. However, to our knowledge, evidence supporting this idea emerged from studies on larvae (including of neotenic species such as the axolotl, *Ambystoma mexicanum*) or juveniles immediately after metamorphosis [10,14]. There is evidence that fully matured frogs form scars on their wounded skin [15,16,17]. Thus, the general rule mentioned above might be true, even for amphibians. Therefore, to develop a model of skin regeneration in the terrestrial life-stage, it is important to find unique animals that do not follow this general rule. While considering these issues, we found that the newt, a kind of urodele amphibian, is an attractive possibility because, among tetrapods, it has exceptional regenerative competency even in its post-metamorphic (or terrestrial) life-stage. Recently, the land-adapted adult axolotl *A. mexicanum*, whose metamorphosis was experimentally induced by injection of thyroid hormone, has been proposed as a good model system [10,18]. However, in cases of salamanders other than the newt, deterioration of regenerative rate and fidelity after metamorphosis must always be considered regardless whether metamorphosis is natural or artificial [19].

Adult newts can regenerate various complex tissues or body parts, including a part of the limbs, tail and jaws as well as that of the brain, heart and eyes (lens and retina) [4]. However, surprisingly, except for limb and tail regeneration, there is limited evidence suggesting that they have the ability to perfectly regenerate their full-thickness skin, although it is widely believed to be so, as supported by a short description of the Iberian ribbed newt, *Pleurodeles waltl*, in a recently published review [20].

Among newt species, the Japanese fire-bellied newt (*Cynops pyrrhogaster*) provides a good model system because a comprehensive transcriptome database, gene manipulation techniques, and animal resources are available [11,21,22,23]. Moreover, there are detailed descriptions for the regeneration of various body parts [5,9,11,12,23,24,25,26,27]. It is noteworthy that in this species, dedifferentiation or reprogramming of terminally differentiated somatic cells plays a role in regeneration during the terrestrial life-stage, particularly in lens [5,26,28] and retina regeneration [8,9,24], and in the creation of muscle [12] and potentially cartilage or bone in limb regeneration [12].

In this study, using *C. pyrrhogaster*, we addressed for the first time whether adult newts can regenerate their full-thickness skin by evaluating the restoration of its texture, appendages and color, while examining the process of re-epithelialization and the extent of granulation or dermal fibrosis. We report that adult newts can regenerate their full-thickness skin very well via unique processes of re-epithelialization and without fibrotic scar formation, except for the dorsal-lateral to abdominal skin whose unique color patterns never recovered. Color pattern is species-specific. Therefore, the adult newt, particularly *C. pyrrhogaster*, provides an ideal model system for cosmetic and clinical studies aimed at perfect skin wound healing and regeneration in postnatal humans.

## 2. Materials and Methods

Experiments presented herein were performed at Keio University, Shinshu University and Tsukuba University. All methods were carried out in accordance with the Regulations for the Handling of Animal Experiments in each university. All experimental protocols were approved by the Animal Care and Use Committee of each university (Keio University, 14090-(1), 14090-(2), 14090-(3); Shinshu University, 300099; University of Tsukuba, 170110). Moreover, all methods were performed in accordance with the ARRIVE guidelines.

### 2.1. Animals

Sexually mature females of the Japanese fire bellied newt *C. pyrrhogaster* were used in this study. They were captured from Niigata, Fukui and Ishikawa prefectures by a supplier (Aqua Grace, Yokohama, Japan), and stored at the University of Tsukuba. The animals were reared in plastic containers, in which the resting place (land) was provided in shallow water, at 18–22 °C under natural light conditions until experiments were performed [21]. For experiments, only healthy animals without visible wounds were selected. Especially, for the long-term evaluation of regenerative competence, animals with almost the same proportion and size (total body length: 12–13 cm; ~90% of fully grown size; estimated age: >5 years) were selectively used. The animals were transported to Keio University and Shinshu University, and then reared in the same conditions. To track the epidermis of the skin, a transgenic adult newt (*CAGGs-mCherry (I-SceI)*) expressing mCherry in a mosaic pattern, was used [12]. Transgenic newts were stored at the University of Tsukuba and all experiments with transgenic newts were carried out at the University of Tsukuba.

### 2.2. Anesthesia

FA100 (4-allyl-2-methoxyphenol; LF28C054; DS Pharma Animal Health, Osaka, Japan) dissolved in water or phosphate-buffered saline (PBS; pH 7.5) were used as anesthetics. Intact newts and newts whose wounds had closed were anesthetized in 0.1% FA100 dissolved in water, in a capped bottle (up to 5 newts per 300 mL) at room temperature (RT: 22 °C) for 45 min to 1 h (before taking pictures) or for 2 hours (before surgical operations) [22]. In the case of newts whose wounds had not closed, they were anesthetized by 10–20 µL abdominal injection of 10% FA100 dissolved in PBS. In these protocols, newts awoke 1–2 h (0.1% FA100, 45 min to 1 h) or 4–6 h after anesthetics (0.1% FA100, 2 h; abdominal injection of 10% FA100), if they were placed at RT. For the end point, the abdomen was injected with a higher dose of FA100 (100 µL injection of 20% FA100).

### 2.3. Surgical Operations

Anesthetized animals were dried on paper towels and placed under a dissecting microscope (M165 FC, Leica Microsystems, Wetzlar, Germany; SZX16, Olympus, Tokyo, Japan). Animals that were anesthetized in water were rinsed in distilled water (DW) before being dried on paper towels. Subsequently, using a microdissection blade, scissors, pin and forceps, a 4–9 mm^2^ square-to-oval shaped piece of full-thickness skin was carefully excised at various locations on the body (head, trunk and limbs; Figures 3 and 4). Note that only one experimental wound was made in each animal to exclude the possible indirect influence by secondary or other wounds. Operated animals were momentarily placed on dry paper towels until the bleeding stopped, then were transferred to moist containers with a lid containing air vents (up to three newts per length 200 mm × width 150 mm × height 55 mm container). Note that the moist container was never filled with water, but was always kept in a semi-dry condition in which the bottom was covered by a moist paper towel after it was tightly wrung, thereby providing folds or bumps on the surface of the crumpled towel that served as resting places for the wounded animals. The skin of animals was allowed to regenerate in the moist containers at RT. Animals were fed daily with frozen *Chironomus* larvae (Akamushi, Kyorin, Hyogo, Japan) [21]. Paper towels were replaced with new ones every other day.

### 2.4. Preparation of Tissue Sections

At desired time points or stages of skin regeneration, animals were sacrificed by excessive injection of the anesthetics (see above section), except for those animals whose forearm or shin was wounded. In those cases, animals were anesthetized by a standard injection method. The body parts containing intact or regenerating skin were separated under a dissecting microscope and immediately transferred into fixatives for the following tissue sectioning. In the case of animals with a wounded forearm or shin, after the limb was amputated in the middle of the upper arm or upper leg (thigh) to collect the samples, the animals were transferred back to moist containers and allowed to regenerate and heal completely for use in other studies.

Cells of the adult newt typically had cell bodies larger than 15 μm in diameter. Therefore, thick frozen sections (20 μm) were used to examine cell types and division and thin paraffin sections (4 μm) were used to examine fine structures of cells and tissues. In the case of frozen sections, samples were harvested into modified Zamboni’s fixative solution (0.2% picric acid/2% paraformaldehyde (PFA) in PBS) on ice, and incubated at 4 °C for 3 h. The samples were washed thoroughly with PBS at 4 °C (15 minutes × 3, 30 min × 2, 1 h), and then allowed to equilibrate in 30% sucrose in PBS at 4 °C. The tissues were embedded in Tissue-Tek^®^ O.C.T. Compound (4583; Sakura^®^ Finetek USA, Inc., Torrance, CA, USA), frozen at about −30 °C in a cryotome (CM1860, Leica Microsystems; CM-41, Sakura, Osaka, Japan), sectioned at about 20 μm thickness, attached on gelatin-coated cover slips, and then dried at RT for a few hours. The sections were stored at −20 °C until use.

In the case of paraffin sections, samples were harvested into 4% PFA in 0.1 M phosphate buffer (PB, pH = 7.4) and incubated at 4 °C for 15 h (for immunohistochemistry, see below). Samples were washed with 0.1 M PB at RT (4 h × 2, then 15 h), trimmed by a surgical knife to create a correct orientation, dehydrated by immersing them in successively higher concentrations of ethanol, transferred to xylene, and finally embedded in paraffin wax (7810, Tissue-Tek^®^ Paraffin Wax II 60, Sakura^®^ Finetek USA, Inc.). Sectioning was performed at a thickness of 4 μm using a rotary microtome (771110, Microm HM355S, Thermo Sci., Walldorf, Germany) with a Section Transfer System (771190, Thermo Sci.). Sections were expanded to their original dimensions by floating them in a warm water bath, and transferred onto silane-coated glass slides (MAS-01, MAS^TM^ Adhesion Microscope Slides, Mastunami Glass, Osaka, Japan). Sections on the slides were air dried on a hotplate at 45 °C for 1 h and then stored at 4 °C until use.

### 2.5. Staining Tissue Sections

Thin paraffin sections were immersed in xylene, rehydrated in successively lower concentrations of ethanol, and then transferred to PBS. Sections were routinely stained with hematoxylin and eosin (Muto Pure Chemicals Co., Ltd., Tokyo, Japan). To visualize collagen-rich extracellular matrix in blue or grey, sections were stained using Masson’s Trichrome system (Muto Pure Chemicals Co., Ltd.) according to the manufacturer’s instructions. Sections (on the glass slide) were embedded in non-aqueous medium containing xylene (107961, Entellan new, Merck Millipore, Burlington, MA, USA) and mounted by a cover slip.

To visualize nuclei and chromosomes under a fluorescence microscope, thick frozen sections were equilibrated to RT, washed thoroughly (PBS, 0.2% TritonX-100 in PBS, PBS; 15 min each), and incubated in DAPI (1:50,000, D1306, Thermo Fisher Scientific, Tokyo, Japan) for 15 h. Sections (on the coverslip) were lightly rinsed in DW, and mounted on a glass slide with 90% glycerol in PBS.

To identify blood cells, thick sections were washed in PBS for 15 min and stained using a standard Wright-Giemsa system (15021, Muto Pure Chemical Co., Ltd.) according to the manufacturer’s instructions. The sections were rinsed twice in M/15 phosphate buffer solution (pH = 6.4, No. 1561-1, Muto Pure Chemical Co., Ltd.) and finally in DW. Sections (on the coverslip) were mounted on a glass slide with 90% glycerol in DW.

For immunohistochemistry, paraffin sections about 4 μm thick were prepared as described above except for the fixation condition in which modified Zamboni’s fixative (see above) was used and fixed samples were rinsed in 70% ethanol for 1 day prior to further dehydration. For antigen retrieval, after sections were deparaffinized and rehydrated as described above, they were incubated in 1× TE buffer (10 mM Tris-HCl, 1 mM EDTA, pH 9.0) at 98 °C for 45 min. Sections were incubated in 3% hydrogen peroxide in DW at RT for 10 min and thoroughly washed in 0.1% Tween-20 containing PBS (PBS-T). Tissues were immunolabelled by the ABC method using the Vectastain *Elite* ABC Rabbit IgG kit (PK-6101; Vector Laboratories, Burlingame, CA, USA) according to manufacturer’s instructions. The primary antibody used in this study was rabbit polyclonal anti-PCNA antibody (1:10,000; ab18197; Abcam plc, Cambridge, UK). Tissues were incubated in blocking solution (PBS-T with goat normal serum) containing biotin (1:50, Avidin/Biotin Blocking kit, SP-2001; Vector Laboratories) at RT for 30 min, and then incubated in primary antibody diluted with blocking solution containing biotin at 4 °C for 15 h. They were washed twice in PBS-T at RT, incubated in biotinylated secondary antibody (affinity-purified anti-rabbit IgG antibody) diluted in blocking solution at RT for 4 h, thoroughly washed in PBS-T, incubated in a mixture of avidin and biotin complex (1:50) for 30 min, then washed twice in PBS-T. The immunoreaction was visualized with the DAB Substrate Kit (SK-4100; Vector Laboratories) according to manufacturer’s instructions. Hematoxylin was used for counter-staining nuclei. Tissues on the glass slide were immersed into malinol (2009-2; Muto Pure Chemicals Co., Ltd.) and covered with a coverslip.

### 2.6. Skin Monitoring, and Image Acquisition and Analysis

Skin regeneration in living animals was monitored using digital camera systems (EOS Kiss x7i, Canon, Tokyo, Japan; DP70, Olympus) attached on dissecting microscopes (M165 FC, Leica Microsystems; SZX16, Olympus). Transmitted light and fluorescence images of tissue sections were acquired by a digital light microscope (BZ-X710, Keyence, Osaka, Japan) or a charge-coupled device (CCD) camera system (DP73, cellSens Standard 1.6, Olympus; DP22, cellSens Standard, Olympus) attached to a fluorescence microscope (BX50; Olympus). Microscopic images were analyzed by Photoshop 21.0.0 (Adobe Systems, San Jose, CA, USA) as well as with software for image acquisition systems. Bands on agarose gels in PCR were scanned and quantified by NIH ImageJ software (http://imagej.nih.gov/ij/ (accessed on 2 February 2020)). Figures were prepared using Photoshop. Image, brightness, contrast, and sharpness were adjusted according to the journal’s guidelines.

### 2.7. Cell Counting

For cell counting, 20 μm thick tissue sections were used. For one animal, four sections across the central region of the wound bed were randomly selected from one forearm sample, the number of cells along the skin or wound epidermis was counted in every section, and finally the values collected from four sections were summed as a representative sample of the animal. The same experiments were repeated at least three times independently. In Figure 4, dividing cells were counted by surveying cells exhibiting mitotic figures. Chromosomes were stained with DAPI and visualized under a fluorescence microscope (see above). In Figure 6, blood cells were stained with the Wright-Giemsa system (see above) and their types were identified according to established criteria [23,29].

### 2.8. PCR Gene Expression Analysis

To investigate gene expression during early skin regeneration, tissues within the wound area, from the surface to a depth of a few millimeters, were collected under a dissecting microscope immediately (stage 1), 6 h, 12 h (stage 2), 18 h, and 96 h (stage 3) after the skin was removed from the forelimb. Tissues from one limb were harvested into one tube. Total RNA was purified from the tissues by Nucleospin RNA (Mini kit for RNA purification; Takara Bio Inc., Shiga, Japan) according to the manufacturer’s instructions except for the amount of rDNase which was 1.5 times higher than that of the recommended protocol. PolyA RNA was converted to cDNA using the SuperScript IV First-Strand Synthesis System (Thermo Fisher Scientific) according to the manufacturer’s instruction.

Using these cDNA samples, semi-quantitative PCR was carried out with the KODFX system (KFX-101, Toyobo, Osaka, Japan) on a T-100 Thermal Cycler (Bio-Rad, Hercules, CA, USA). The genes examined in this study and the primer sets and cycle numbers applied to PCR are listed in Appendix A, respectively. Following electrophoresis, the amount of PCR products was quantified by NIH ImageJ software. *EF1α* was used as the internal control [23]. The relative change in gene expression during regeneration was calculated by normalizing the expression level of the gene at each stage by that of *EF1α* at the same stage and dividing the normalized expression level of the gene at each stage by the maximum value during regeneration. At each stage, cDNA samples were prepared more than three times, i.e., from three independent limbs.

### 2.9. Data Analysis

Data in the text are presented as the mean  ±  SE. For cell counting data, the statistical difference between samples was evaluated by Shirley-Williams’ multiple comparison test by using BellCurve for Excel (ver 3.22, Social Survey Research Information, Tokyo, Japan). For PCR data, statistical differences between means were evaluated by a multiple comparison test (Tukey HSD method) following ANOVA using IBM SPSS Statistics 23 (IBM, Chicago, IL, USA).

## 3. Results and Discussion

### 3.1. Fundamental Structure of the Normal Skin in Adult C. pyrrhogaster

We examined the structure of normal skin in the adult newt, *C. pyrrhogaster*. Regardless of the location on the surface of the body, the adult newt skin was, as in other vertebrates, composed of the epidermis and the dermis (Figure 1a). The epidermis was further stratified with three layers corresponding to the basal layer, transitional layer and stratum corneum (Figure 1b). The basal layer is the innermost layer comprised of cuboidal or columnar stem cells, referred to as ‘basal stem cells’ in this paper. The transitional layer is where cells produced from the basal layer gradually mature while being pushed up toward the stratum corneum. The stratum corneum is the outermost layer formed by fully-matured squamous cells, and is sometimes shed off as a result of renewal (or metabolic turnover) of the epidermis. In histological sections, we occasionally observed mitotic figures along the basal layer (Figure 1c–f). However, interestingly, cell division took place not only vertically toward the stratum corneum but also horizontally along the plane of the basal layer. These observations suggest that the epidermis in the adult newt continuously grows while renewing itself.

Thus, the fundamental structure of normal skin in the adult newt was similar to that in humans. However, the thickness of the epidermis in the adult newt was typically 3–5 cells, much thinner than that in humans (30–50 cells thick) [30]. In humans, the transitional layer has a thickness of 13–18 cells and can be subdivided into three layers: the stratum spinosum, stratum granulosum and stratum lucidium [30]. However, in the adult newt, the thickness of the transitional layer varied between one and three cells, and was occasionally missing in some histological sections (Figure 1a).

*C. pyrrhogaster* is one type of fire-bellied newt, and has a black and orange color pattern on the skin of its belly. An accurate description, at least for the animals used in this study, is as follows: the dorsal skin of the head, trunk, limbs and tail is uniformly dark (brown to black), and the ventral skin and skin around the border between the dorsal skin and ventral skin (defined as ‘the lateral skin’ here) in these body parts had a black and orange pattern. However, a light yellow or golden area was occasionally observed, but it was not examined in this study. Note that the color pattern in this species is individually unique. In dorsal skin, two kinds of pigment cells were identified. One was localized in the epidermis (Figure 1g,h). It had a small cell body in the innermost layer of the transitional layer, and extended dendritic fibers along the border between the basal layer and the transitional layer, as if the fibers enfolded basal stem cells. We named it the ‘dendritic melanophore’ here. The other one was an ordinary melanophore, referred to simply as ‘melanophore’ here, which resided along the uppermost region of the dermis, forming the pigment cell layer (Figure 1b). In addition to dendritic melanophores, epidermal cells in the dorsal skin also contained melanin pigments in their cytoplasm, making the epidermis brown (Figure 1b,g). The melanin pigments in the epidermal cells may originate from melanophores. In humans, epidermal cells also contain melanin pigments that originate from melanocytes, which are sparsely distributed along the basal layer [31].

In the lateral-to-ventral skin with a black and orange color pattern, another kind of orange pigment cell was discovered, the ‘xanthophore’ (Figure 1i,j). Xanthophores lay along the same level of melanophores, forming a pigment cell layer in the orange skin area. Melanophores and xanthophores did not form distinct layers. Along the border of black and orange areas, membranous margins of melanophores and xanthophores were intertwined (Figure 2a). Note that the margins of xanthophores sometimes extended to slightly deeper levels than those of melanophores (Figure 2b). This could explain observations under a dissecting microscope, where melanophores at the border of black and orange areas appeared to overlie the sheet of xanthophores. When focus was turned to the epidermis, in contrast to opaque dorsal skin, the epidermis was transparent (Figure 1i,j and Figure 2a,b). Epidermal cells did not contain melanin pigments or dendritic melanophores, even in black areas. On the other hand, in ventral skin, fibrous tissue layers containing melanin pigments lay along the dermis (Figure 2a).

The dermis of the adult newt was, as in other vertebrates, a collagen-rich connective tissue with appendages such as exocrine glands as well as pigment cells (i.e., melanophores and xanthophores) (Figure 1a). Granular glands and mucous glands were recognized as exocrine glands. Similar to sweat glands and sebaceous glands in humans, exocrine glands in amphibians are believed to have differentiated from invaginations of the basal layer [32]. In amphibians, the cutaneous respiration system is highly developed. In the adult newt, blood capillaries for cutaneous respiration formed a horizontal network embedded at almost the same level of the pigment cell layer, and these were observed in between exocrine glands in histological sections (Figure 1g–j). Unlike humans, in the adult newt, a subcutaneous tissue layer containing adipocytes (i.e., a fat layer) was not recognized (Figure 1a and Figure 2a).

### 3.2. Adult C. pyrrhogaster Can Pass Almost All Test Items to Be Evaluated for Skin Regeneration in Reconstructive Surgery and Cosmetic Studies

We investigated the regenerative capacity of adult newt skin. We removed a 4–9 mm^2^ square-to-oval shaped piece of full-thickness skin at three locations on the body (head, trunk and limbs) and then monitored changes to the wound for over two years (Figure 3 and Figure 4; also see Appendix A). We used sexually mature female newts of almost the same size because the adult newt expresses sexual dimorphism and changes its body proportion as it grows [33]. After excision of the full-thickness skin, muscle epimysium (in the case of trunk and limbs) or bone periosteum (in the case of dorsal skin of the head) was exposed due to the absence of a subcutaneous tissue layer (Day 0, Figure 4). Note that in the case of abdominal skin, muscle and peritoneum were also wounded because attachment of the skin and muscle was too tight to be separated (Figure 4i). In postnatal humans, such severe injuries to the skin definitely give rise to granulation followed by dermal fibrosis, leading to a scar, where skin texture, appendages and color hardly recover [2]. In contrast, the adult newt regenerated skin very well. A summary of this evaluation is shown in Figure 3b.

In this evaluation, regenerating skin at different locations was evaluated at 180 days after operation regardless of whether the wound was covered by the epidermis (re-epithelialization), whether granulation or dermal fibrosis took place, or whether the texture (patterned sulcuses or grooves), appendages (e.g., granular and mucous glands) and color (hue/tone and pattern) were restored. For this evaluation, the structure and color of regenerating skin of anesthetized animals were examined under a dissecting microscope. Tissue samples containing the regenerating skin were then fixed and sectioned to examine dermal fibrosis and appendage restoration under an optical microscope. In all samples at 180 days after operation, the wound bed of the skin had re-epithelialized completely (100%) but there was no indication of either granulation or dermal fibrosis (0%). However, texture and color (tone) in the regenerating area did not always recovered sufficiently to blend into the surroundings (Day 180, Figure 4). However, in the long-term monitoring over 2 years, which was performed independently of the 180 day experiments, we found that the texture and color (tone) in the dorsal and ventral skin eventually recovered at 720 days after operation (*: 100%, *n* = 2; see below). Exceptionally, animals from which a plain area of the ventral skin was excised unfortunately died during long-term monitoring. Therefore, to evaluate the capacity of ventral skin to restore color (tone), we used animals from which a patterned area was excised, and eventually found that the color tone of the orange area almost reached the levels of its surroundings at 720 days after operation (**: 100%, *n* = 2; see below).

In the case of appendages, we judged their restoration by examining in histological sections whether new glands appeared or not, regardless of their maturation. At 180 days after operation, all samples had immature or grown glands (100%). The density and size of glands almost reached the levels of their surroundings by 720 days after operation. By this stage, granular and mucous glands had become identifiable (see below).

Taken together, adult newt skin could regenerate very well regardless of its location on the body, with the exception of the dorsal-lateral to ventral skin and the cloaca skin, whose color pattern hardly recovered (see following sections).

### 3.3. Fundamental Processes of Skin Regeneration in Adult C. pyrrhogaster

We carried out histology of different days of regenerating skin at various locations in a long-term evaluation. Finally, we found that the fundamental processes of skin regeneration were common among locations. Based on these observations, we defined eight morphological stages of regenerating skin (Figure 5).

When the full-thickness skin was excised (Stage 1), re-epithelialization started with a multilayered epidermis (referred to as ‘wound epidermis’ here) which extended from the wound margin (Stage 2; 6–12 h after operation), and completed as the wound epidermis covered the wound bed within 1–3 days (Stage 3). Following re-epithelialization, dermal reconstruction started: a collagen-rich dermal layer (yellow) gradually thickened from the wound margin (Stage 4; 4–7 days after operation); slightly after the beginning of this event, melanophores or xanthophores at the wound margin started to migrate to restore the pigment cell layer under the wound epidermis (Stage 5; 7–14 days after operation); exocrine glands started to differentiate from the basal layer of the wound epidermis within 2–4 months after operation (Stage 6). Following the appearance of exocrine glands, the regenerating skin started its maturation: between 3 months and 6 months after operation, the thickness of the dermal layer had almost reached the levels of its surroundings, and the pigment cell layer had been reformed throughout the area of the wound epidermis, although the exocrine glands were still premature (Stage 7). Eventually, between 1 year and 2 years after operation, the regenerating skin had almost blended into the surroundings (Stage 8). A representative data set obtained from the dorsal skin of the forearms is shown in Figure 6.

In conclusion, the fundamental processes of skin regeneration in the adult newt were similar to those described in larval and land-adapted salamanders [10,18], and frogs immediately after metamorphosis (called ‘juvenile frogs’ here) [14], suggesting that the master plan for skin regeneration is conserved in amphibians. Subsequently, we further characterized the processes of skin regeneration in the adult newt.

### 3.4. Adult C. pyrrhogaster Re-Epithelializes the Wound Bed in a Unique Manner

We examined the early processes of skin regeneration in greater detail, with a focus on re-epithelialization. To achieve this, we used the dorsal skin of forearms because the forearms of the adult newt had been subjected to the study of limb regeneration and the skin on the dorsal side is less vulnerable to animal movement [12,23,27].

#### 3.4.1. The Fundamental Processes of Re-Epithelialization

When full-thickness skin was excised, a multilayered wound epidermis appeared at the margin of the wound within 6 h (Figure 7a,b). Cells of the basal layer and those in the lower part of the transitional layer extended together to form the wound epidermis while leaving the upper part of the transitional layer and the stratum corneum behind. In fact, the extending epidermal structure and the other part of the epidermis frequently detached during tissue sectioning, suggesting that their attachment is physically weak (Figure 7c,d). In addition, the wound epidermis carried dendritic melanophores inside, which should have originally been between the basal layer and the transitional layer at the wound margin (Figure 7e–h; for comparison, see Figure 1g). Importantly, the leading edge of the early wound epidermis is generally believed to be in the form of one cell layer, is flat or thin and attached to the substrate, and tows the trailing body of multilayered wound epidermis [34]. In contrast, in the adult newt, such a structure could not be recognized at the leading edge of the wound epidermis throughout re-epithelialization; instead, a multicellular structure with a streamlined shape was always observed in histological sections (Figure 7b,e). This unique structure (named the ‘leading end’ here) of the wound epidermis probably formed by reorganization of the cells of the basal layer and those in the lower part of the transitional layer immediately after the full-thickness skin was excised. The thickness of the leading end obviously depended on the thickness of the wound edge of the skin (Figure 7b,e).

As the wound epidermis extended, its appearance under a dissecting microscope became tongue-like (Figure 7i). Therefore, as in other papers [33], we termed the wound epidermis between Stages 2 and 3 as the ‘extending epidermal tongue (EET)’. The EET had a leading end that was sometimes followed by a thinner epithelium (2–3 cells thick) comprised of the basal layer and the transitional layer. Interestingly, the EET, which was brown, preserved melanin pigments in most constituent cells for as long as the wound bed had been covered. Since migration of ordinary melanophores from the wound margin began after the wound had closed (Stage 4–5), the presence of melanin pigments in the EET is thought to be a carry-over since the newly created epithelium is added to the proximal end of the EET (see below). In addition, these observations suggest that turnover of the wound epidermis is very slow during its extension. In fact, differentiation of the stratum corneum started from the circumference of the wound, later than the extension of the wound epidermis, and proceeded as a wave toward the center, as if the stratum corneum covered the wound epidermis (Figure 7i). The extending stratum corneum caught up to the leading end of the EET by the time the EET met together to close the wound (Figure 7j). Finally, the laminar structure and thickness of the wound epidermis matured, which resembling a normal epidermis, even though the development of exocrine glands had not begun (Stage 3).

#### 3.4.2. Cell Sources for Extension of the Wound Epidermis

We assessed which cells had contributed to the extension of the wound epidermis. To discover dividing cells that created the wound epidermis, we first used PCNA-immunohistochemistry in transverse sections of the forearms undergoing re-epithelialization. Unexpectedly, almost all basal stem cells were labelled by PCNA antibody, even in intact skin, making it difficult to evaluate changes in proliferative activity along the basal layer (Figure 8). In the newt, it has been suggested that PCNA is not specific to the S-phase of the cell cycle but is expressed throughout the cell cycle [24,35].

Fortunately, since the newt’s chromosomes are large enough to recognize in tissue sections (see Figure 1c–f) [36], it was possible to survey cells exhibiting mitotic figures to evaluate the probability of cell division. In the area of the wound bed (WE in Figure 9a), cell division was never observed along the EET (Figure 9b). This observation was consistent with those in other vertebrates, in which the basal stem cells slightly apart from the wound margin may be a dominant source of the wound epidermis [10,34,37,38]. Therefore, we assumed that cell division should be accelerated in the skin near the wound margin. Importantly, consistent with observations in other vertebrates [10,34,37,38], dividing cells in the skin surrounding the wound (in the case of the transverse section corresponding to the Lateral and Ventral areas in Figure 9a) were, as in normal skin, the only cells in the basal layer (i.e., the basal stem cells) of the epidermis.

We first examined the area 300 μm behind the wound margin. Mitotic figures were occasionally recognized along the basal layer, and positions of their spindle poles suggested that both vertical and horizontal cell division had taken place in this range (Figure 9c–f). However, no clues could be found regarding the location of dividing cells, nor were there significant changes in the number of dividing cells during re-epithelialization. Therefore, the range was expanded to the other side of the forearm (i.e., the Lateral area in Figure 7a), discovering that the number of dividing cells significantly increased to about two-fold as early as 6 h after operation, declined as re-epithelialization proceeded, and then returned to the level of normal skin (Stage 1) as the wound closed (Stage 3) (Figure 9b). These observations suggest that a large area of the skin surrounding the wound was activated upon excision of full-thickness skin, where the probability of cell division along the basal layer was overall higher (about two-fold, on average), creating cells for the extension of the wound epidermis. It is possible that the basal stem cells produced cells horizontally as well as vertically, that these cells became integrated into the original basal layer and transitional layer, and then a subset of cells in both layers were displaced laterally toward the wound bed (see below).

Interestingly, basal stem cells in the wound epidermis never displayed cell division once the wound epidermis had exited the wound margin. However, these cells re-started cell division once the wound epidermis had covered the wound bed completely (Stage 3) (Figure 9g–i). This should contribute to restoration of the structure of the epidermis and re-start of its self-renewal (metabolic turnover), although there was no significant difference in the probability of cell division in the basal layer between the wound epidermis at Stage 3 and the epidermis in normal skin (cumulative number of dividing cells in four sections (mean ± SE, *n* = 3): epidermis in the normal skin, 3.3 ± 1.0 cells/mm; wound epidermis at Stage 3, 4.0 ± 2.2 cells/mm).

To test the hypothesis that the wound epidermis is not derived from basal stem cells localized along the wound margin but instead from the epidermis that has expanded from a large area around the wound, a skin tracking study was conducted using transgenic adult newts (Figure 10). We first tried to graft a piece of fluorescent skin of a transgenic individual onto the corresponding region of a wild-type or albino individual. The grafted skin survived on the host until blood flow had recovered in it, but it degraded slowly thereafter, possibly due to immunological rejection [27,39]. Therefore, the plan was switched to using mosaic individuals whose back skin had fluorescent spots, although the number of these animals were very limited. We carefully selected one particular individual (*CAGGs > mCherry (I-SceI)* [12]) in which a cobble stone-shaped epithelial cell sheet (i.e., the stratum corneum) was observed on the surface of spots by its mCherry fluorescence (Figure 10a). This indicated that basal stem cells in the epidermis of the spot had been recombined, enabling us to monitor the movement of the epidermis along the wound edge during extension of the wound epidermis. We excised full-thickness skin from the center of the fluorescent spot and tracked the fluorescent skin remaining along the wound edge (Figure 10c–e). As a result, the fluorescent epidermis shifted its location towards the center of the wound. Interestingly, in contrast to the epidermis, the glands and texture around the wound were almost unchanged, as were their positions and patterns. These observations support our hypothesis that skin wounds are closed by an overall expansion of the epidermis over a large area around the wound, rather than by an epidermis created from a narrow area of the epidermis along the wound edge. To confirm the results of skin tracking, it is necessary to increase the number of available individuals in the future.

#### 3.4.3. Wound Surface on Which the Wound Epidermis Extends

The scaffold which allowed the wound epidermis to attach and extend on the surface of the wound bed was examined. It is generally believed that coagulation following bleeding on the wound serves as a fibrin matrix and as an initial scaffold for the wound epidermis [10,34,40]. In the adult newt, when full-thickness skin was excised from the dorsal surface of the forearm, slight bleeding took place from capillaries along the pigment cell layer at the wound margin, and occasionally from the wound bed in which the muscle layer (extensor muscle) was composed of bundles of muscle fibers and in which blood capillaries were also wounded (Figure 4c). In most cases where bleeding from the wound bed was minimum, only a thin line of blood clot formed immediately along the circumference of the wound bed, contributing to prompt hemostasis. In these cases, on the wound bed, the surface (epimysium) of the muscle layer was temporarily exposed to air, although it was soon covered by a thin layer of blood clot and tissue fluid (Figure 4c). Therefore, during surgical operation, the surface of the muscle layer suffered varying degrees of damage between animals (Figure 11a,b): in some animals, most of the epimysium was not left behind (Figure 11a). Three hours after operation, the surface of the wound bed was covered by a fibrin-like membrane which resembled a solid cover sheet under a dissecting microscope (Figure 11c). It is possible that, by that time, the structure of the fibrin matrix had been altered to a non-elastic form [41]. Interestingly, by 6 h after operation, regardless of the degree of damage, the surface of the muscle layer consistently became smooth with connective tissue (Figure 11d). This suggests that the surface of the wound bed, or the epimysium, was quickly repaired before the wound epidermis had arrived. However, unexpectedly, the leading end of the wound epidermis never met the wound bed until it arrived at the wound surface at around 12 h after operation (Figure 11e; also see Figure 7b,e and Figure 9c). Thus, the fibrin-like membrane and epimysium tissue seemed unlikely to be the primary substrate for the wound epidermis, but rather a coat to protect the tissues under the wound.

After the leading end of the wound epidermis arrived at the wound surface, the wound epidermis (or EET) extended further over the fibrin-like membrane or epimysium tissue (Figure 12). However, importantly, there was a gap between the EET and fibrin-like membrane or epimysium tissue. The gap was composed of tissue fluid containing white blood cells (Figure 12b). This finding suggested that adhesion between the wound epidermis and the wound bed might be weak. In fact, the leading end of the EET was often detached from the wound bed in histological sections (Figure 12a). We speculate that such a weak attachment of the leading end to the wound bed may be responsible for the lack of a flat/thin leading edge on the EET. Unlike other vertebrates, the wound epidermis of the adult newt might extend as if it glides over the wound bed. The tissue fluid on the surface of the wound bed may work as a lubricant.

Taken together, re-epithelialization in the adult newt skin was unique. A schematic diagram summarizing the results is shown in Appendix A.

Besides, re-epithelialization in the adult newt skin also seemed to be unique compared to that in its limb regeneration. In limb regeneration in amphibians, it is believed that the wound surface of the amputated limb is promptly covered by a thin epithelial layer (i.e., wound epidermis) of flattened cells that migrated from the basal layer at the wound margin, and that the epithelium thickens to become multilayered as the wound surface is covered [10,37,38]. The wound epidermis in the central region develops into an apical epithelial cap (AEC) which is thicker than a normal epithelium [10,37,38]. In contrast, as shown in this study, the wound epidermis in adult newt skin regeneration was multilayered from the beginning to the end of re-epithelialization, and was created from a large area of the skin surrounding the wound. Additionally, the region corresponding to AEC in the wound epidermis could not be identified. Probably, cellular mechanisms and regulations underlying formation and extension of the wound epidermis may be different between skin regeneration and limb regeneration. Alternatively, there may be a possible difference in the developmental stage and/or species because the concepts related to the formation of the wound epidermis during limb regeneration were derived from studies of larval salamanders and juvenile frogs [10,37,38]. Importantly, it seems that the land-adapted adult axolotl also has a similar shape of the wound epidermis during skin regeneration [18]. It is essential to clarify these hypotheses in the adult newt by cell-tracking strategies in combination with molecular and cellular biology in a future follow-up study.

### 3.5. Inflammation in Adult C. pyrrhogaster Never Induces Granulation and Dermal Fibrosis

Four-limbed vertebrates in the aquatic life-stage are generally endowed with the competence of skin regeneration without a scar (termed ‘scarless wound healing’), but this competence declines or is lost as they transit to the terrestrial life-stage [6,10,15,16,17]. Therefore, a striking feature of adult newt skin regeneration is undoubtedly scarless wound healing. In this study, we attempted to confirm this competence in adult *C. pyrrhogaster*. Furthermore, we carefully compared the processes associated with skin wound healing between the adult newt and postnatal human from the perspective of scar formation. In postnatal humans, following hemostasis, the processes associated with skin wound healing were divided into three phases (inflammatory, proliferation and remodeling phases), which overlap with each other [10,34,42]. A prolonged inflammatory reaction on the wound bed leads the formation of granulation tissue followed by fibrotic scar formation [2,17,43].

In postnatal humans, hemostasis is thought to be an essential event for the following inflammatory reaction [2,10,42]. Activated platelets release chemokines that recruit neutrophils and then macrophages, and these initiate and sustain the inflammatory response. Following the inflammatory phase, neutrophils and macrophages release cytokines and chemokine to recruit various kinds of cells (lymphocytes, fibroblastic cells, endothelial cells, and keratinocytes) that contribute to the next proliferation phase. In the proliferation phase, lymphocytes and other immune cells continue the protective response initiated by neutrophils. Fibroblastic cells in the surrounding tissues migrate to the wound bed and then proliferate, while secreting collagen, fibrinogen and other ECM proteins, and these replace the fibrin matrix (blood clot), forming an initial scaffold for re-epithelialization. This event brings about the development of granulation tissue, which is believed to be imperative for proper tissue repair and angiogenesis. The formation of granulation tissue is accompanied by the extension of new blood capillaries into the interior, and these support the growth of the granulation tissue. During the formation of granulation tissue, the wound epidermis continues to extend on the second scaffold provided by the granulation tissue, and eventually re-epithelialization is completed. As the wound closes, a scar is formed from the wound margin where fibroblastic cells (myofibroblasts) contract while depositing collagen-rich ECM. The proliferation phase is followed by a remodeling phase where neutrophils, macrophages and endothelial cells (excessive blood capillaries) finally disappear, leaving collagen-rich ECM in the wound bed (dermal fibrosis). The physical strength of the scar is determined by the properties of the epidermis, as well as the quality and quantity of collagen in the wound bed, and those are regulated by fibroblastic cells.

#### 3.5.1. Inflammatory Reaction on the Wound Bed

By contrast, in the adult newt, structures corresponding to either granulation or dermal fibrosis could not be found, even though an initial inflammatory reaction was detected in the wound bed (Figure 13). Dilation of blood capillaries, a sign of inflammation, was not evident in the wound bed after hemostasis. However, at 18 h after operation when the EET had covered about two-thirds of the area of the wound bed, the number of mesenchymal cells—including neutrophils—obviously increased in the wound bed (Figure 13a–d,p). This was remarkable in the space under the epimysium tissue in the residual one-third of the area where the wound surface was still exposed to the external environment (Figure 13a–d). Therefore, changes in the number of mesenchymal cells at a depth of 200 μm under the epimysium tissue in the wound bed was investigated during re-epithelialization (Figure 13e,f). In this study, red blood cells (matured erythrocytes only), eosinophils, mast cells (or basophils), neutrophils and others (i.e., an unidentified fraction) were identified according to their characteristic morphology and staining properties (Wright-Giemsa stain) [23,29] (Figure 13g–p). It must be noted that the unidentified fraction contained immature erythrocytes, thrombocytes, monocytes (macrophages), and lymphocytes. However, it was difficult to count these cells accurately because their shape and color were too ambiguous in this experimental condition. Additionally, it was difficult to exclude the possible presence of mesenchymal cells other than blood cells, such as fibroblastic cells around the epimysium or satellite cells around muscle fibers. At any rate, after excision of full-thickness skin, the number of mesenchymal cells in the wound bed gradually increased, reached a maximum as the re-epithelialization reached two-thirds of completion (18 h), and then declined to the normal level as re-epithelialization became completed (2–3 days; Stage 3).

The kinetics of identified immune cells was analyzed. The number of eosinophils seemed to increase between 6 h and Stage 2 (12 h after operation), and then remain stable at least until Stage 3 (Figure 13g). The number of mast cells (or basophils), which were frequently observed even in normal tissue, increased abruptly between 6 h and Stage 2 (12 h), and then returned to normal levels in the following 6 h (Figure 13j). On the other hand, the number of neutrophils started to increase immediately after operation but remained at relatively low levels until Stage 2 (12 h). Their levels increased drastically between Stage 2 (12 h) and 18 h, and then decreased, reaching initial levels (Figure 13m). Collectively, even though these types of immune cells displayed different kinetics in response to the excision of full-thickness skin, it could be concluded that the inflammatory reaction in the wound bed was initiated as early as 6 h after the excision of full-thickness skin. Similar to postnatal humans, blood coagulation might have caused the recruitment of immune cells as an initial phase of the inflammatory reaction. Granulocytes and mast cells are generally believed to secrete chemokines or cytokines, further recruiting other kinds of immune cells as well as themselves during a second phase of the inflammatory reaction [44]. The dramatic increase of neutrophils at 18 h might have been a result of the second phase of the inflammatory reaction.

Changes in the relative quantity of transcripts of inflammatory cytokines (IL-1β, IL-6, IFN-γ and iNOS) [16,45,46,47,48,49,50,51], and those of anti-inflammatory cytokines (TGFβ, IL-10 and Arg1) [46,47,48,51,52] in the wound bed undergoing re-epithelialization (Figure 14; Appendix A) were investigated. Overall, the quantity of transcripts of inflammatory cytokines examined did not change obviously, except for IL-6, whose quantity of transcript gradually increased, reaching a significantly higher level at 18 h after operation. Note that IFN-γ was not detected in either intact skin, intact subcutaneous tissue or the wound bed during re-epithelialization (Appendix A). Interestingly, in both IL-6 and iNOS, the quantity of transcripts declined significantly as the wound closed completely (Stage 3). On the other hand, the anti-inflammatory cytokines that were examined tended to increase upon operation, peaking at 18 h, and then decreasing as the wound closed completely (Stage 3). For both TGFβ and Arg1, the levels at 18 h in the wound bed undergoing re-epithelialization were significantly higher than those in intact subcutaneous tissue.

These results suggest that in the adult newt, moderate levels of inflammation and suppression of inflammation may occur synchronously when full-thickness skin is excised, and both return to their original state when the wound is closed. The fact that both responses peaked at 18 h after operation is interesting because it corresponds to the peak accumulation of mesenchymal cells, including neutrophils, in the wound bed (Figure 13f,m). In the future, it will be necessary to identify more immune cells, including lymphocytes and monocytes (macrophages), and to investigate their dynamics and involvement in inflammation in relation to cytokine expression.

Collectively, these observations suggest that skin regeneration in the adult newt also involves an inflammatory phase. However, based on the kinetics of mesenchymal cells and the expression of cytokine genes in the wound bed, it is suggested that inflammatory reactions ceased overall soon after re-epithelialization was complete (see Appendix A).

#### 3.5.2. Dermal Reconstruction without Scarring

Importantly, as described above, re-epithelialization in the adult newt proceeded during the inflammatory phase. Moreover, structures corresponding to the granulation tissue were never found under the EET during this phase, indicating that unlike postnatal humans, granulation is not required for re-epithelialization in the adult newt.

In this study, we described the processes of dermal reconstruction following re-epithelialization. After completing re-epithelialization, collagen-rich dermal tissue began to expand from the circumference of the wound (Stage 4, Figure 5 and Figure 6). It is reasonable to hypothesize that migration of fibroblastic cells from the surroundings of the wound might have been triggered by inflammatory reactions in the wound bed, and that fibroblastic cells might have participated in reformation of the dermal structure by depositing collagen-rich ECM while proliferating in the space between the wound bed and the wound epidermis. This process might correspond to the proliferation phase in postnatal humans [10]. However, unlike postnatal humans, this phase was never accompanied by the formation of granulation tissue, but instead by reformation of a new dermal tissue furnished with a pigment cell layer, blood capillaries for cutaneous respiration, and exocrine glands.

As the collagen-rich dermal tissue expanded along the space between the muscle layer and the wound epidermis, melanophores in the pigment cell layer at the wound margin started to migrate along the space between the collagen-rich dermal tissue and the wound epidermis (Stage 5, Figure 5 and Figure 6). Blood capillaries also extended from the wound margin along with extension of the pigment cell layer (Figure 15). The presumptive dermal layer thickened while being equipped with exocrine glands (Stage 6–7, Figure 5 and Figure 6), and eventually acquired almost the same structure as that in normal skin (Stage 8, Figure 5 and Figure 6).

It is necessary to note that the time required to complete dermal reconstruction varied widely among individuals, especially the period from Stages 6 to 8, which took between 4 and 21 months (Figure 5). This period might correspond to the remodeling phase in humans [10]. However, the appearance or disappearance of blood capillaries in the collagen-rich dermal tissue between the pigment cell layer and muscle layer were never observed. Moreover, characteristics of dermal fibrosis, such as an altered pattern of the collagen matrix or excessive deposition of ECM or similar properties of the scar, such as contraction and hardening of the skin, were never recognized [53].

During the regeneration of amputated limbs in the adult newt, the wound also closes promptly while skipping the formation of granulation tissue [12,23]. However, in this case, a large number of immune cells such as monocytes (macrophages) gather in the space between the wound bed and wound epidermis after the wound has closed [23]. This event is followed by formation of the blastema, which is comprised of mesenchymal cells including potentially reprogrammed/dedifferentiated cells [12,54,55] originating from dermal fibroblastic cells that migrated from tissues proximal to the amputation plane. A large number of neutrophils are observed in this growing blastema, toward which neovascularization occurs [23]. Thus, blastemal tissue formation appears to partially or fully implement the biological processes common to those required for the formation of granulation tissue. However, unlike granulation tissue, blastemal tissue is patterned to reform a complex structure of the missing part of a limb [27].

In this study, no indications of blastemal tissue formation throughout the processes of skin regeneration were ever found. In skin regeneration, re-epithelialization seemed to be directly followed by reconstruction of the dermal structure, whereas in limb regeneration, it must be followed by more complicated processes that orchestrate the creation and patterning of constituent tissues (bone, muscle, nerve, blood or lymphatic vessels) as well as the skin [12,23,27]. In fact, even though dermal cells of the adult newt skin are capable of regenerating cartilage or bone during limb regeneration [12], such potency was never manifested during skin regeneration. Thus, it seems that the potency of dermal cells is not released in the context of skin regeneration. However, as was observed in a study of larval amphibians and amphibians immediately after metamorphosis (juvenile amphibians) [14], it still needs to be clarified whether dermal re-construction in skin regeneration and blastema formation in limb regeneration share common mechanisms to recruit mesenchymal cells from surrounding tissues. Importantly, neither fibrosis nor scar formation were observed in either skin or limb regeneration. It is interesting, as was suggested in retinal regeneration of the adult newt [11], that myofibroblasts, which are the major contributor to the progress of fibrosis and scar formation in postnatal humans [53,56], are not created under the control of a reprogramming/dedifferentiation mechanism. In a future follow-up study, these issues must be addressed while determining the cellular origin of the new dermis in skin regeneration of the adult newt.

### 3.6. Adult C. pyrrhogaster Can Restore Its Skin Texture and Appendages

One of the demands of cosmetic or surgical operations is the recovery of skin texture and appendages. In adult *C. pyrrhogaster*, a patterned structure on the surface of the skin was recognized (Figure 16a–d). This structure had similar characteristics to those of skin texture in postnatal humans and was comprised of small bumps, each of which was surrounded by sulcus and mostly had an exocrine gland (granular or mucous gland) on the inside. Therefore, here we defined the bump and sulcus on the skin as the newt’s crista cutis and sulcus cutis, respectively. In addition, grooves on the lateral skin of the trunk were recognized (Figure 16e–i), and these might be a derivative of costal grooves in the larval stage [57].

The appearance and recovery of skin texture after the wound closed was traced. We found that skin texture eventually recovered so efficiently that the wound blended into its surroundings, although the time taken for recovery differed among individuals (see Figure 3 and Figure 5). In some cases, the crista cutis and grooves were restored within 180 days after operation, becoming almost identical to those of intact skin (Figure 4a–f,i,j and Figure 16e–g). However, in other cases, it took as long as two years (Figure 16h,i). Such individuality was observed even in the same region of the body (for example, compare Figure 4c,d with Figure 16j).

During skin regeneration, as the area of collagen deposition in the presumptive dermal layer expanded toward the center of the wound (between Stages 4 and 5; Figure 5 and Figure 6), the sulcus cutis or grooves first appeared in the epidermis. As the wound region became entirely dark while the pigment cell layer underneath the epidermis was reformed (Stage 6; Figure 5 and Figure 6), a pattern of the sulcus cutis became obvious, although the crista cutis was still flat (Figure 16h). The tissue section at this stage revealed that the differentiation of exocrine glands from the basal layer of the epidermis just started under the region of the presumptive crista cutis (Figure 16j). As the exocrine glands grew while the dermal layer thickened to a normal level, the crista cutis also grew, increasing in both size and thickness (Stage 7; Figure 5 and Figure 6). Finally, the structural pattern of the skin surface in the wounded area integrated into that of its surroundings (Figure 16i). At this stage, granular and mucous glands were separately identifiable based on their morphological characteristics (Stage 8; Figure 5 and Figure 6). Taken together, it can be concluded that adult newts could restore skin texture as well as skin appendages, although the time necessary for restoration varied among individuals. It is believed that the interaction between the epidermis and dermis is essential for the differentiation of skin texture and appendages [6,58]. This might also be true for skin regeneration in the adult newt. In fact, Stages 6 to 8, which correspond to the period needed to complete the new dermis, showed large variation among individuals.

### 3.7. Adult C. pyrrhogaster Can Restore the Color Hue and Tone of Skin, but Not Its Unique Color Pattern

The recovery of skin color after an operation is also a very important test item that needs to be confirmed in cosmetic and clinical studies. In adult *C. pyrrhogaster*, most of the dorsal skin is brown-black and its ventral skin is orange with a unique black pattern. However, it must be noted that the color pattern on the belly is different among individuals, even in the same regional race [59]. Moreover, in some individuals, orange dorsal skin near the border with the ventral skin (dorsal-lateral) was spotted (Figure 16e and Figure 17a). Skin color resulting in brown-black and orange are primarily caused by melanophores and xanthophores, respectively.

The changes in color (hue or tone and pattern) of the wound region were traced after full-thickness skin was excised (Figure 17). In the head, limbs and trunk, the brown-black dorsal skin eventually recovered its color hue and tone, although the time necessary for the recovery of tone varied (between 180 days and 2 years) among individuals, as was observed for skin texture (see Figure 3 and Figure 16). On the other hand, when a part of the skin was excised across a color pattern, the pattern appeared by 180 days after operation, but the original pattern never reformed in either the dorsal-lateral or ventral skin (Figure 16e and Figure 17a–d). In all cases, the size of the orange area on skin decreased or its shape changed because melanophores migrated to the wound bed quicker than xanthophores and expanded the black area or created new black areas, finally altering color pattern. In ventral skin, when a part of the plain orange area was excised without wounding the black area, the color hue in the wound region recovered by 180 days after operation (Figure 17e). In this case, there was collective migration of xanthophores whereas melanophores hardly migrated (Figure 17f,g). However, the color tone of the orange area had not recovered by 180 days after operation in either the plane area (Figure 17e) or patterned area, even in cases of a patterned area where the color tone of the black area had recovered (Figure 17h,i). Long-term monitoring revealed that it took as long as two years for the orange area to recover its color tone (Figure 17h,i).

Taken together, it can be concluded that adult newts have the capacity to restore the hue and tone of skin color, even after full-thickness skin was excised, although the time needed for tone to recover varied among individuals. Considering this variability, restoration of the color tone of skin seemed to be related to individual speed of dermal reconstruction, as was observed in the restoration of skin texture and appendages. The time to restore color tone in the orange area seemed to be longer than that in the black area. This may be due to differences in the requirements for pigment synthesis between melanophores and xanthophores, for example, a difference in substrates that need to be supplied from foods, which may be insufficient in the current captive-rearing conditions [60].

On the other hand, the adult newt was obviously incompetent in reforming its original color pattern in the dorsal-lateral and ventral skin. Of note, skin texture and appendages were restored, even in conditions where color pattern was deformed (Figure 16). This suggests that in adult newt skin regeneration, the regulation of structural pattern may be independent of the regulation of color pattern, which is determined by the patterning of melanophores on two-dimensional coordinates of the skin. To our knowledge, there are few studies in *C. pyrrhogaster* that have addressed developmental mechanisms or physiological functions of color patterning in their belly skin. Color pattern in the belly displays large variation among individuals, even in the same regional race [59]. Therefore, it would be difficult to assume developmental mechanisms such the creation of a species-specific fixed-pattern in skin color. Moreover, it would be difficult to assume the physiological signals for intra-species communications, even though orange on the belly skin is believed to serve as a warning against predators [61]. Independently, from cosmetic and clinical points of view, such imperfect restoration of the color pattern in skin would not be satisfactory for the newt. Consequently, we must conclude that the adult newt could not perfectly regenerate its skin.

## 4. Conclusions

In this study, an attempt was made to identify a reliable animal model to study skin regeneration in postnatal humans, i.e., humans in the terrestrial life-stage. To achieve this, the competence of adult newts to regenerate their skin were evaluated for the first time using cosmetic and clinical perspectives. Using the Japanese fire-bellied newt *C. pyrrhogaster*, full-thickness skin was excised at various locations on the body, and re-epithelialization, granulation or dermal fibrosis, and recovery of texture, appendages, and color (hue, tone and pattern) were examined. After a long-term evaluation (>2 years), it was discovered that adult newts could eventually effectively regenerate their skin, regardless of its location on the body, except for dorsal-lateral and ventral skin whose original color pattern was never restored. Note that the concern about color pattern is specific to newts because humans do not usually have such color patterns in skin. Therefore, excluding the aspect of color pattern, adult *C. pyrrhogaster* could serve as an ideal model of skin regeneration in postnatal humans.

In this study, morphological changes of the skin wound during regeneration was further investigated, and the processes of skin regeneration were successfully divided into eight stages (Figure 5). Moreover, we confirmed that the adult newt provides a scarless wound healing model. It was also discovered that wounds closed quickly (within a few days) in the adult newt, with the remaining period dedicated to skin regeneration (as long as 2 years) and dermal reconstruction. It makes sense to protect the wound bed from infection by pathogens by closing the wound as soon as possible, and to carefully deal with the following complicated tasks related to regeneration without disturbance by inflammatory or immune responses. In fact, the initial inflammatory reaction on the wound bed was terminated soon after the wound had closed. It is generally believed that granulation followed by dermal fibrosis is brought about as a result of a prolonged inflammatory reaction on the wound bed [17,43,53]. As predicted, in the adult newt, neither granulation nor dermal fibrosis (or fibrotic scar) were ever recognized in regenerating and regenerated skin. Thus, rapid re-epithelialization seems to be a key to skipping granulation, leading to scar-free skin. The adult newt controlled re-epithelialization uniquely (see Appendix A). At minimum, the following mechanisms could contribute to the speeding up of re-epithelialization: (i) supplying regenerative materials (cells for both the basal layer and transitional layer) to the wound epidermis from a large area of the skin surrounding the wound; (ii) providing a soft substrate on the wound bed enabling the wound epidermis to glide over the wound bed smoothly; (iii) enabling the wound epidermis to continue its extension, regardless of the systemic inflammatory or immune response. Another key for scarless wound healing could be dedifferentiation or reprogramming. Tetrapods commonly acquire the ability to heal a wound with fibrotic scars while losing their high regenerative competency in the aquatic life-stage as they transit to a terrestrial life-stage [4,6,7,8,10]. Importantly, a recent study of retinal regeneration in adult *C. pyrrhogaster* proposed that dedifferentiation or reprogramming, which are unique to the adult newt, may have evolved from a mechanism underlying fibrosis or scar formation [11]. It is worthy to hypothesize that fibroblastic cells involved in the secretion of collagen-rich extracellular matrix may undergo a unique process of dedifferentiation or reprogramming that turns their fate from the formation of granulation tissue (i.e., scar formation) toward dermal reconstruction (i.e., scarless regeneration). In a future study, these two keys need to be thoroughly analyzed at molecular and cellular levels to obtain clues that would allow the regenerative strategy of the adult newt to be mimicked and extrapolated to scarless skin wound healing in human medicine.

## Figures and Tables

**Figure 1 biomedicines-09-01892-f001:**
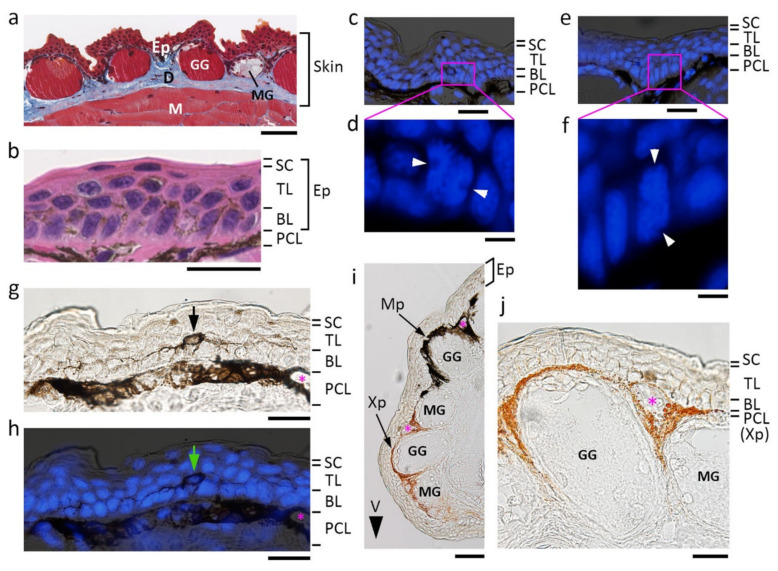
Fundamental structure of the adult newt skin. Images were obtained from transverse sections of the dorsal skin of the forearm unless otherwise mentioned. (**a**) A representative image of the skin. Masson’s trichrome stain. Blue/gray area: collagen-rich connective tissue. (**b**) A representative image of the epidermis. Hematoxylin-eosin stain. Cells in both the basal layer and the transitional layer contained melanin pigments. (**c**–**f**) Sample images showing cell division along the basal layer. DAPI stain of nuclei or chromosomes. White arrowheads: spindle pole. The basal stem cells divided either horizontally (**c**) or vertically (**e**). (**g**,**h**) Representative images of the dendritic melanophore. DAPI stain of nuclei is shown in (**h**). The cell had a cell body (arrows) located at the innermost region of the transitional layer, and extended dendritic fibers along the border between the transitional layer and the basal layer. (**i**) A representative image of lateral skin with a black and orange color pattern. V: ventral side. (**j**) A representative image of the plane orange area of ventral skin. The top is the ventral side. Note that the epidermis was transparent in lateral and ventral skin with a color pattern. We could not detect dendritic melanophores or pigment granules in cells of the basal and transitional layers. BL: basal layer; D: dermis; Ep: epidermis; GG: granular gland; M: muscle; MG: mucous gland; Mp: melanophore; PLC: pigment cell layer; SC: stratum corneum; TL: transitional layer; Xp: xanthophore. Asterisks: blood capillaries. Scale bars: 100 μm (**a**,**i**); 40 μm (**b**,**g**,**h**,**j**); 50 μm (**c**,**e**); 10 μm (**d**,**f**).

**Figure 2 biomedicines-09-01892-f002:**
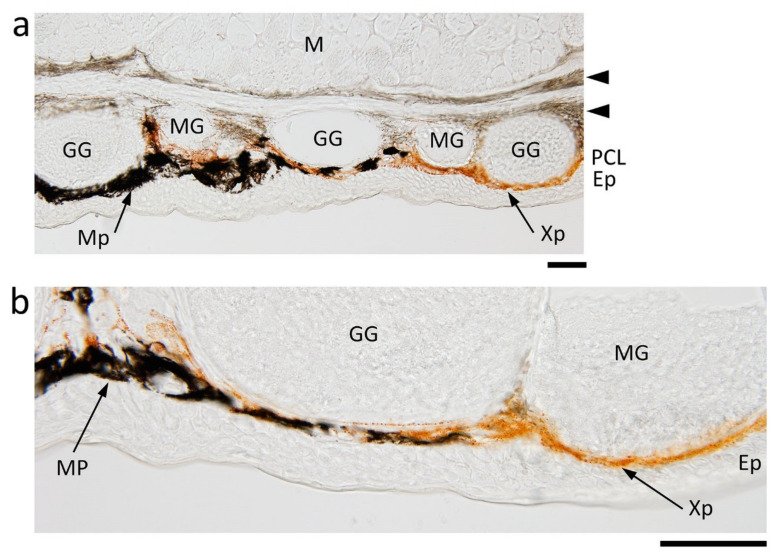
Abdominal skin of the adult newt. (**a**,**b**) Representative images of transverse sections of the border of black and orange areas. Ep: epidermis; GG: granular gland; M: muscle; MG: mucous gland; Mp: melanophore; PLC: pigment cell layer; Xp: xanthophore. Arrow heads: pigmented fiber layers. Scale bars: 100 μm.

**Figure 3 biomedicines-09-01892-f003:**
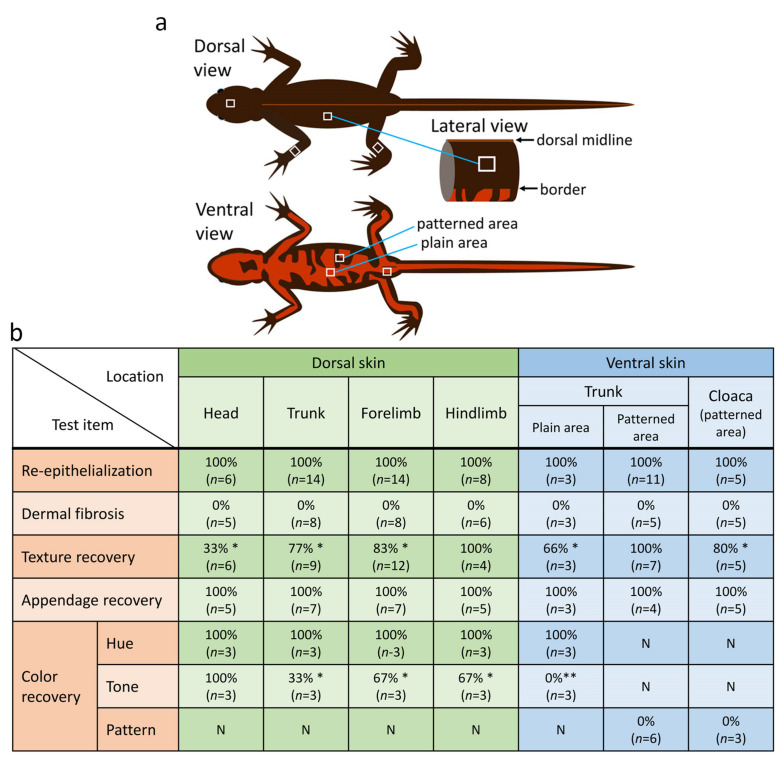
An evaluation of regenerative capacity of skin at various locations on the body of the adult newt. (**a**) Schematic drawing of the site of skin removal. A 4–9 mm^2^ square-to-oval shaped piece was excised from the dorsal skin of the head, trunk and limbs (forelimbs and hind limbs), as well as from abdominal skin, including skin around the cloaca (white squares). In the dorsal skin of the trunk, an excision was made within the area between the top (dorsal midline) and the boundary with orange abdominal skin. In the abdominal skin (i.e., belly skin), the plain orange area and the black-orange patterned area were separately examined. (**b**) Summary of results at 180 days after operation. * and ** show the results at 720 days after operation; each was 100% (*n* = 2) (for details, see the main text). N: not applicable.

**Figure 4 biomedicines-09-01892-f004:**
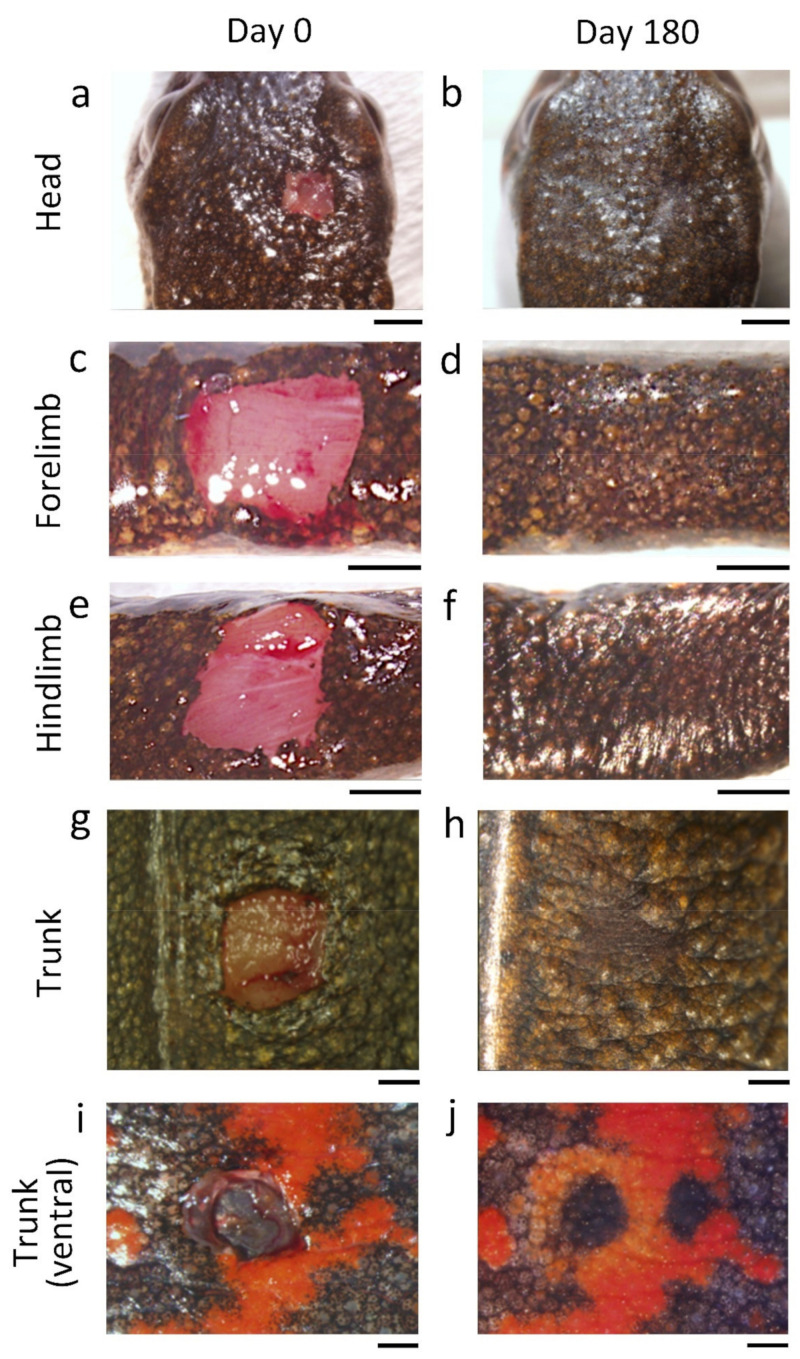
Sample images of regenerating skin at different locations on the body of the adult newt. The left- and right-hand columns show the images immediately (Day 0) and at 180 days (Day 180) after operation, respectively. (**a**,**b**) Dorsal skin of the head. (**c**,**d**) Dorsal skin of the forearm. (**e**,**f**) Dorsal skin of the shin of a hind limb. In these three cases, regenerating area had almost blended into the surroundings by 180 days. The conditions in (**b**,**d**,**f**) were almost acceptable according to the requirements of reconstructive surgery and cosmetic studies. (**g**,**h**) Dorsal skin of the trunk. In this case, the area near the dorsal midline was excised (see Figure 3a). Note that the texture and color tone in this area had not fully recovered by 180 days. (**i**,**j**) Color patterned area of the ventral skin of the trunk (i.e., the belly skin). In this particular case, the shape of the black area was altered although the skin structure had recovered well by 180 days. Note that the orange area surrounding the black area had not been restored to its original tone by 180 days. The conditions in (**h**,**j**) were not acceptable from a reconstructive surgical or cosmetic standpoint. For these animals, the wound healing process continued to be monitored for as long as 720 days after surgery (see following sections). Scale bars: 2 mm (**a**,**b**); 1 mm (**c**–**j**).

**Figure 5 biomedicines-09-01892-f005:**
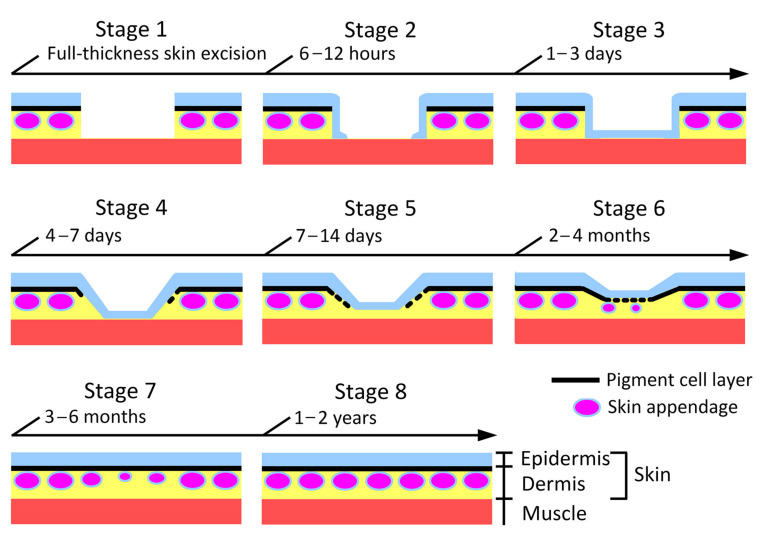
Morphological stages of skin regeneration in the adult newt.

**Figure 6 biomedicines-09-01892-f006:**
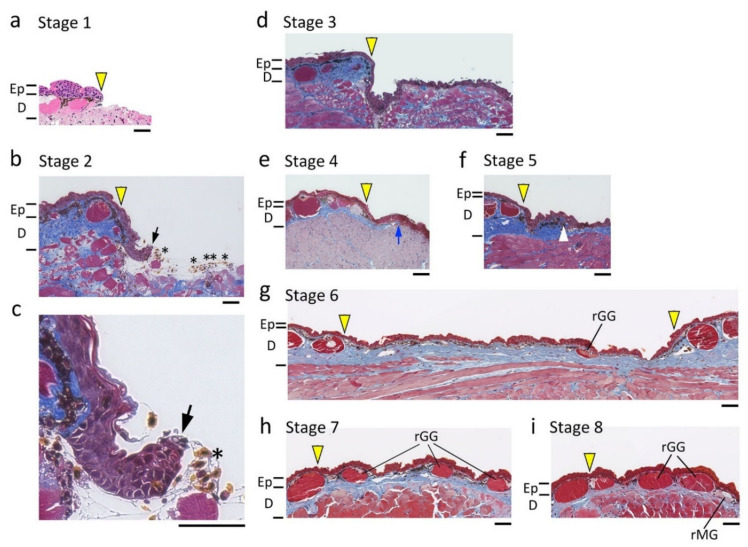
Morphological changes of the skin wound during its regeneration in the adult newt. A representative data set was obtained from the dorsal skin of the forearms. The full-thickness skin was excised from the dorsal surface as in Figure 4c, and allowed to regenerate for as long as 2 years. Images were obtained from transverse sections. Masson’s trichrome stain was used, unless mentioned otherwise. Blue or gray: collagen-rich extracellular matrix (ECM). (**a**) Stage 1: Immediately after the excision of full-thickness skin. Hematoxylin-eosin stain. (**b**,**c**) Stage 2: 12 h after operation. The wound epidermis appeared on the wound bed. Black arrow: leading end of the wound epidermis. Note that it was multilayered. Asterisks: coagulated blood; red blood cells are dark yellow. (**d**) Stage 3: 3 days after operation. The wound bed was completely covered by the wound epidermis. (**e**) Stage 4: 4 days after operation. The collagen-rich dermal tissue obviously started to expand from the wound margin. Blue arrow: top of the collagen-rich dermal tissue. (**f**) Stage 5: 7 days after operation. Melanophores obviously started to migrate from the wound margin. White arrowhead: top of the migrating melanophores. By this stage, the space in between the wound bed and wound epidermis was entirely filled by collagen-rich dermal tissue, as suggested by the blue or gray color by Masson’s trichrome stain. (**g**) Stage 6: 120 days after operation. Exocrine glands appeared. In this section, a granular gland was recognized. Repetition of the thinner region corresponding to the sulcus cutis became recognizable along the regenerating epidermis. (**h**) Stage 7: 180 days after operation. Exocrine glands grew. By this stage, the collagen-rich dermal tissue thickened to the level of the surrounding, and the pigment cell layer was restored. (**i**) Stage 8: 720 days after operation. The skin regenerated. The regenerating skin had almost blended into its surroundings. By this stage, different types of exocrine glands differentiated, and the crista cutis was also restored. In this section, a mucous gland as well as a granular gland were recognized. Yellow arrowheads: location of wound margin. D: dermis; Ep: epidermis; rGG: regenerating granular gland; rMG: regenerating mucous gland. Scale bars, 100 μm.

**Figure 7 biomedicines-09-01892-f007:**
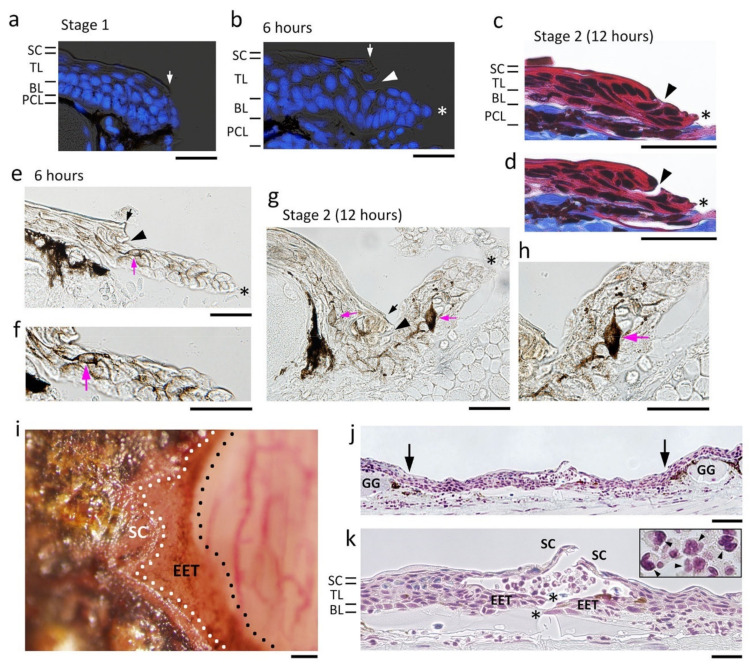
The process of re-epithelialization. A representative data set was obtained from the dorsal skin of the forearms. (**a**) Wound margin immediately after operation (Stage 1). (**b**) Wound epidermis which appeared 6 h after operation. Blue in (**a**,**b**): DAPI staining of nuclei. (**c**,**d**) Neighboring sections showing the wound epidermis at Stage 2 (12 h after operation). Masson’s trichrome stain. As shown in (**d**), the region composed of the stratum corneum and a part of the transitional layer was sometimes detached from the extending wound epidermis during sectioning. (**e**–**h**) Dendritic melanophores (magenta arrow) in the extending wound epidermis. (**e**,**f**) 6 h after operation. (**f**) is a magnification of a part of (**e**). (**g**,**h**) Stage 2 (12 h after operation). (**h**) is a magnification of a part of (**g**). (**i**) The extending epidermal tongue (EET) 6 h after operation. A view from above. The EET was brown because its constituent cells contained melanin pigments. Following a delay after the wound epidermis started to extend, differentiation of the stratum corneum began from the circumference of the wound. (**j**,**k**) Wound epidermis closing around the center of the wound bed 24 h after operation. (**k**) is a magnification of a part of (**j**). This is a sagittal section of the forearm. The wave of differentiation of the stratum corneum caught up with the closure of the wound epidermis; however, in this particular case, the stratum corneum and the EET were not integrated into one continuous epithelium. The central hole surrounded by the wound epidermis was filled with white blood cells containing neutrophils and monocytes (arrowheads in the inset). BL: basal layer; GG: granular gland; PCL: pigment cell layer; SC: stratum corneum; TL: transitional layer. Black and white arrows: location of the wound margin. Arrowheads: margin of the stratum corneum and a part of the transitional layer that were left behind the wound epidermis. Asterisks: leading end of the wound epidermis. Scale bars, 50 μm (**a**–**h**,**k**); 100 μm (**i**,**j**).

**Figure 8 biomedicines-09-01892-f008:**
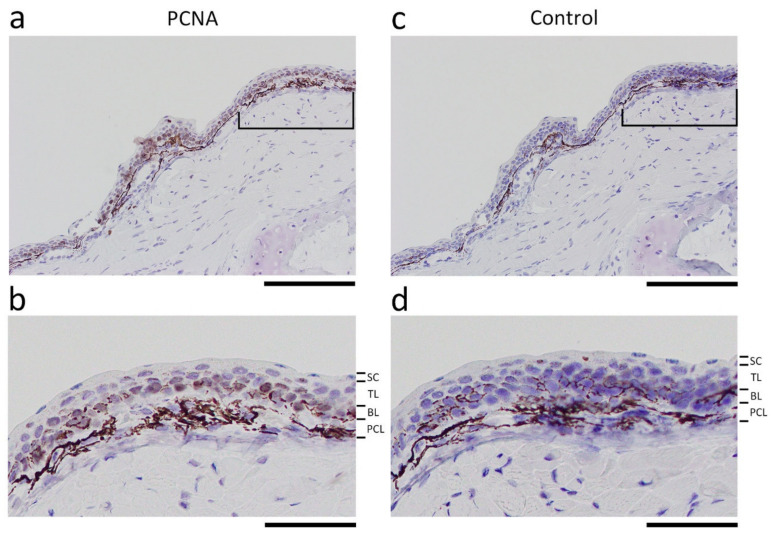
PCNA immunoreactivity in intact skin. (**a**) A representative image showing PCNA immunoreactivity in a transverse section of an intact forelimb. (**b**) Enlargement of the skin in the area indicated by the bracket in (**a**). Almost all the cell nuclei along the basal layer (BL) showed immunoreactivity (light brown). Nuclei in the tissue were counterstained purple with hematoxylin. (**c**) Control. A section next to the one shown in (**a**) was stained as in (**a**), but without primary antibody. (**d**) Enlargement of the skin in the area indicated by the bracket in (**c**). PCL: pigment cell layer; SC: stratum corneum; TL: transitional layer. Scale bars: 300 μm (**a**,**c**); 100 μm (**b**,**d**).

**Figure 9 biomedicines-09-01892-f009:**
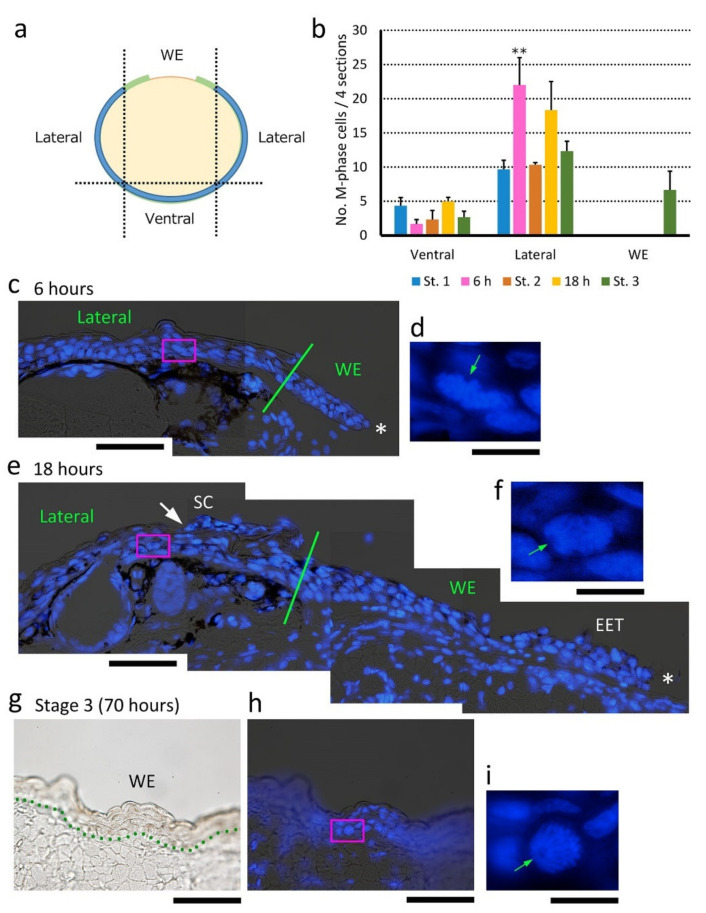
Cell sources for extension of the wound epidermis. (**a**) A schematic drawing of a transverse section of a forearm, illustrating the terms of the regions in the skin (WE, Lateral and Ventral). In WE, the wound epidermis/extending epidermal tongue (EET) from both sides (light green) was examined. (**b**) Changes in the number of M-phase cells in defined regions during re-epithelialization. Forearms at Stage 1 (immediately after operation), 6 h, Stage 2 (12 h), 18 h, and Stage 3 (48–70 h) were examined. For Lateral, the number of cells on both sides of the forearm were summed. Data were collected from four sections per forearm, and the data obtained from three forearms (three newts) were compared between stages (see Methods). In Lateral, the number of M-phase cells, or dividing basal stem cells, significantly increased at 6 h (Shirley-Williams’ multiple comparison test, **: *p* < 0.025), and then returned to a normal level as re-epithelialization proceeded. The wound epidermis started cell division as soon as re-epithelialization was completed (Stage 3). (**c**–**f**) Sample images showing mitotic figures in Lateral at 6 and 18 h. The boxes in (**c**,**e**) were enlarged in (**d**,**f**), respectively. Green lines: position of the wound margin. Asterisks: leading end of the wound epidermis and the EET. The stratum corneum (SC) in (**e**) was folded (the white arrow points to its distal end). Green arrowheads: spindle pole. The cells in (**d**,**f**) were dividing vertically and horizontally, respectively. In (**d**), two sets of daughter chromosomes had slid horizontally, probably because this cell got caught in collective cell migration. It must be noted that observation of dividing cells in this region was occasional. (**g**–**i**) Sample transmitted and DAPI images showing a mitotic figure in the closed wound epidermis (Stage 3). Seventy hours after operation. The dotted line in (**g**) indicates the border between the wound epidermis and the wound bed. The box in (**h**) was enlarged in (**i**). Scale bars: 100 μm (**c**,**e**,**g**,**h**); 20 μm (**d**,**f**,**i**).

**Figure 10 biomedicines-09-01892-f010:**
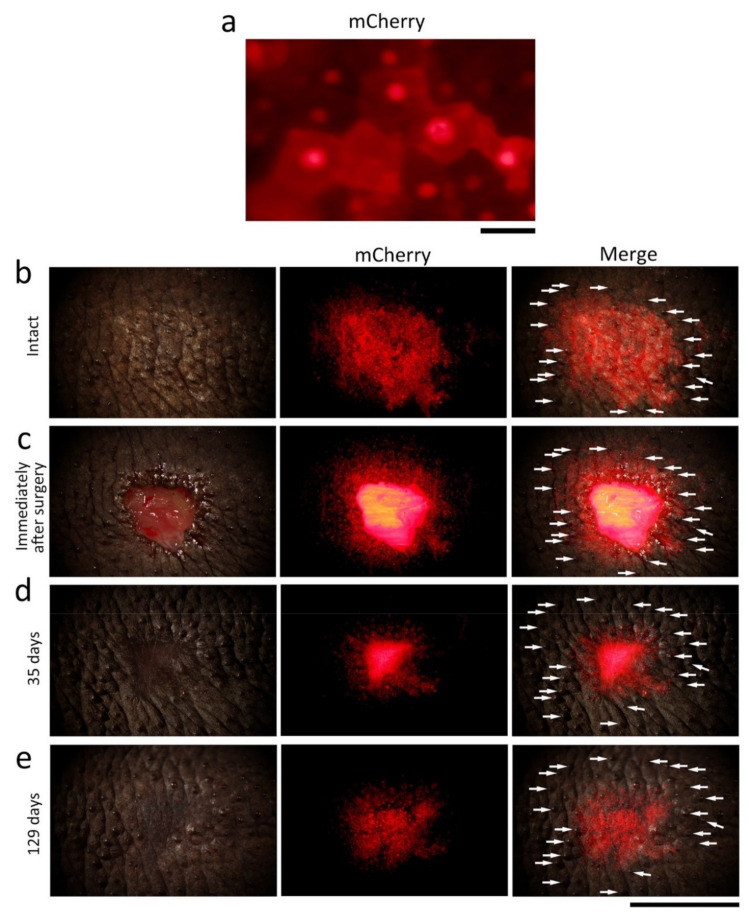
Tracking of the epidermis around the wound margin during skin regeneration. A mosaic pattern of a transgenic adult newt (*CAGGs > mCherry (I-SceI*)) was used in which a red fluorescent protein, mCherry, was expressed in spots on the back skin. (**a**) A magnified view of the stratum corneum in the fluorescent spot under a dissecting microscope. The mCherry fluorescence of cobble stone-shaped epithelial cells made them visible. (**b**–**e**) Tracking of the fluorescent epidermis. A central part of the fluorescent spot was excised and then the fluorescent epidermis around the wound was monitored. (**b**) Before surgery (intact). (**c**) Immediately after surgery. The wound bed fluoresced because muscles intensely expressed mCherry in this animal. (**d**) 35 days after surgery. (**e**) 129 days after surgery. Left-hand column: bright light image. Central column: mCherry fluorescence. Right-hand column: Merge. Arrows in the right-hand column indicate secretion glands as landmarks. We waited for the stratum corneum and the upper part of the transitional layer of the fluorescent epidermis to be renewed (the stratum corneum was shed every two to three weeks) allowing the fluorescent area of the basal layer and the lower part of the transitional layer to be tracked, both of which should slide toward the wound bed according to our hypothesis. Consistently, at 35 days after surgery, the fluorescent area had shrunk compared to the landmarks. The fluorescence of the wound epidermis became recognizable as the wound bed was covered by the pigment cell layer, which blocked off fluorescence from the muscles (129 days). At 129 days, the fluorescent area seemed to have expanded slightly, probably because the skin had relaxed. Note that this experiment was conducted on a single newt, so the same experiment needs to be repeated on multiple individuals to corroborate the results. Scale bars: 40 μm (**a**); 5 mm (**b**–**e**).

**Figure 11 biomedicines-09-01892-f011:**
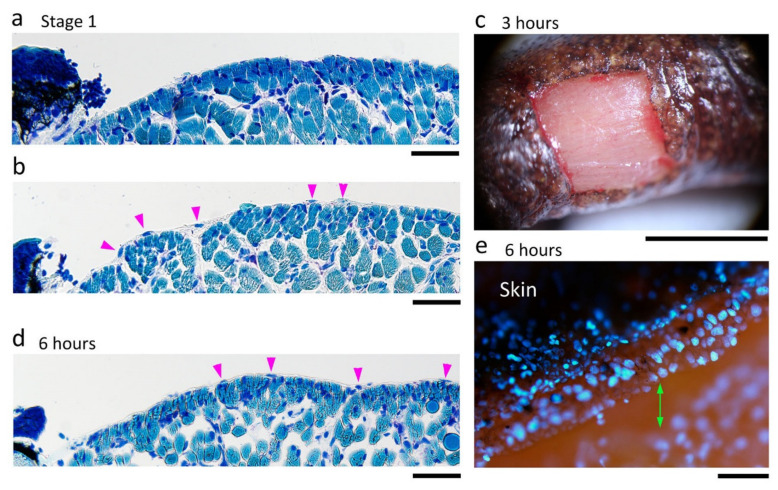
Wound surface after the excision of full-thickness skin from the dorsal surface of the forearms. (**a**,**b**) Sample images showing the wound surface (i.e., the muscle layer) immediately after operation (Stage 1). These images were obtained from different animals. Wright-Giemsa stain. In this staining condition, cell nuclei were stained in dark blue, and muscle fibers were in light blue. On the other hand, connective tissues and blood capillaries were not stained. Therefore, the thickness of connective tissues or kinds of blood cells could be examined under Nomarski optics. In (**a**), the connective tissue (the epimysium) forming the surface of the muscle layer was mostly removed together with the skin, and therefore the muscle fibers were sometimes exposed to air. In (**b**), the epimysium remained. Arrowheads point to the nuclei of cells embedded in the epimysium. (**c**) Sample image of the wound surface at 3 h after operation. View under a dissecting microscope. Slight bleeding occurred along the wound margin but the blood immediately coagulated, leading to hemostasis. A thin membrane-like structure (possibly a fibrin membrane) covered the surface of the wound bed. (**d**) Representative image of the wound surface at 6 h after operation. In all 48 sections obtained from three forearms (16 sections each) whose wound surface suffered varying degrees of damage, the wound surface at 6 h was, as shown here, always smooth with the epimysium tissue, suggesting that the surface of the muscle layer repaired itself very quickly. In the repaired epimysium, nuclei were sometimes recognizable (arrowheads). (**e**) A view of the early wound epidermis (6 h after operation) from diagonally above. Cell nuclei were visualized by DAPI fluorescence. The wound epidermis at this stage extended, but was never attached, to the surface of the wound bed (double arrow). Scale bars: 100 μm (**a**,**b**,**d**,**e**); 2 mm (**c**).

**Figure 12 biomedicines-09-01892-f012:**
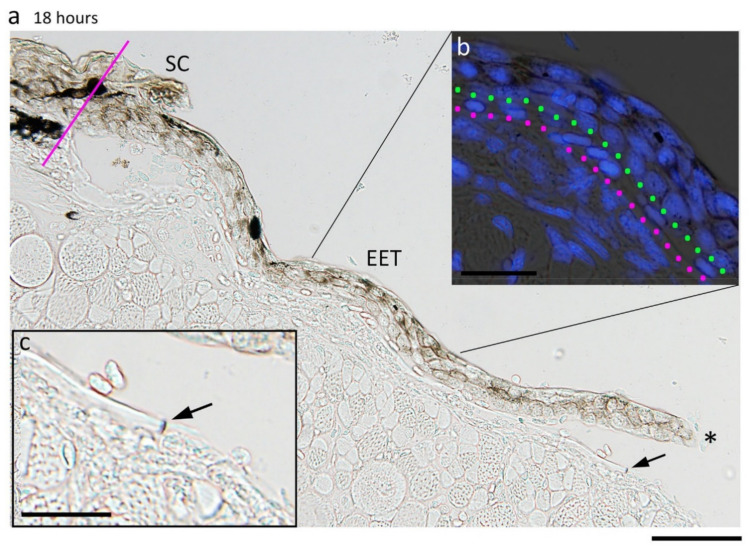
Wound surface on which the wound epidermis extends. (**a**) Representative image showing the wound surface under an extending epidermal tongue (EET) at 18 h after operation. Asterisk: leading end. SC: stratum corneum. Line: position of the wound margin. The EET lay over either the fibrin-like membrane or the epimysium tissue with a space containing white blood cells. A part of the EET is enlarged in inset (**b**). Blue: DAPI stain of nuclei. Dotted line in green: the innermost margin of the EET. Dotted line in magenta: fibrin-like membrane. Nuclei of white blood cells were recognized on the fibrin-like membrane. A break of the fibrin membrane near the leading end of the EET, as indicated by an arrow in (**a**), is enlarged in inset **c**. Scale bars: 50 μm (**a**); 20 μm (**b**,**c**).

**Figure 13 biomedicines-09-01892-f013:**
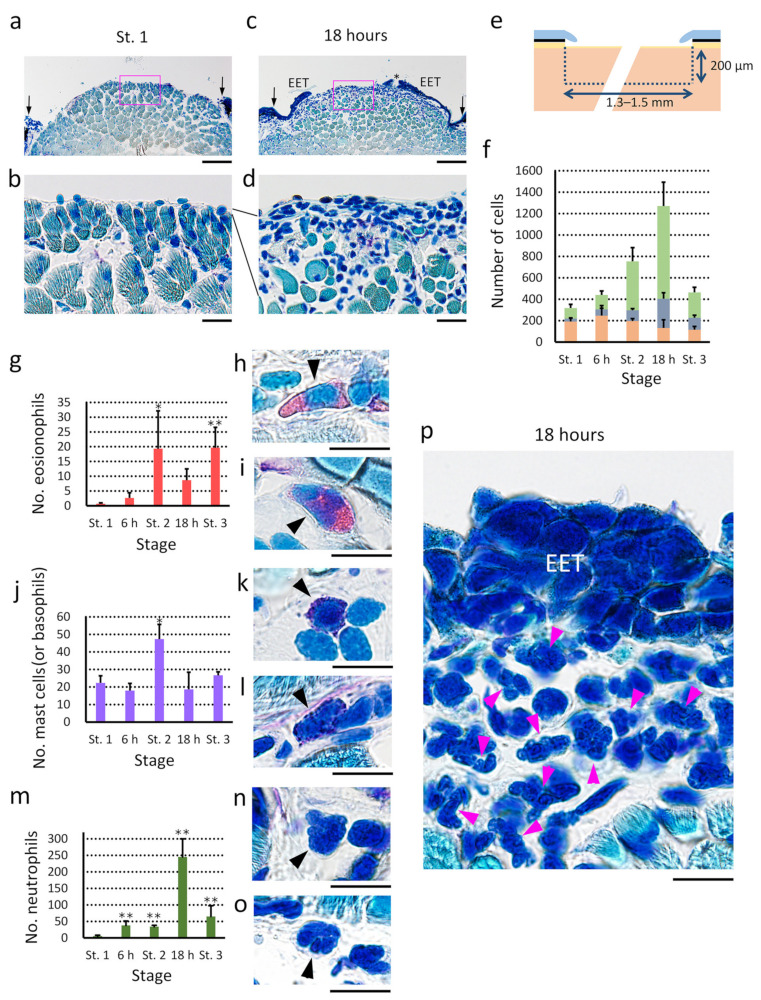
Inflammatory reaction in the wound bed. Representative images were obtained from transverse sections of the forearms at Stage 1 (immediately after operation), 6 h, Stage 2 (12 h), 18 h, and Stage 3 (48–70 h). Wright-Giemsa stain. (**a**–**d**) Mesenchymal cells in the wound bed at Stage 1 and 18 h. The boxes in (**a**,**c**) were enlarged in (**b**,**d**), respectively. EET: extending epidermal tongue. Arrows in (**a**,**c**): position of the wound margin. Dotted lines between (**b**,**d**) show the increase of the space between the wound surface (epimysium) and muscle layer at 18 h, in which mesenchymal cells obviously gathered. (**e**) A schematic drawing of a section, illustrating the region in which the number of mesenchymal cells was counted. (**f**) Changes in the number of mesenchymal cells in the defined region during re-epithelialization. Data were collected from four sections per forearm, and the data obtained from three forearms (three newts) were compared between stages (see Methods). EM(B)N: eosinophils, mast cells (or basophils) and neutrophils; RBC: mature erythrocytes. (**g**) Changes in the number of eosinophils. (**h**,**i**) Sample images of eosinophils. The cells characteristically had cytoplasm stained pink or orange, and sometimes had a lobated nucleus. (**j**) Changes in the number of mast cells (basophils). (**k**,**l**) Sample images of mast cells (basophils). The cells characteristically had granules stained in dark purple in their cytoplasm. Note that in this study mast cells and basophils could not be differentiated. (**m**) Changes in the number of neutrophils. (**n**,**o**) Sample images of neutrophils. The cells characteristically had a multi-lobed nucleus. (**p**) Sample image of neutrophils (arrowheads) gathering in the wound bed at 18 h after operation. In this study, statistical significance of the increase against the value at Stage 1 was analyzed by Shirley-Williams’ multiple comparison test (**: *p* < 0.025; *: *p* < 0.05). Scale bars: 200 μm (**a**,**c**); 40 μm (**b**,**d**); 20 μm (**h**,**i**,**k**,**l**,**n**–**p**).

**Figure 14 biomedicines-09-01892-f014:**
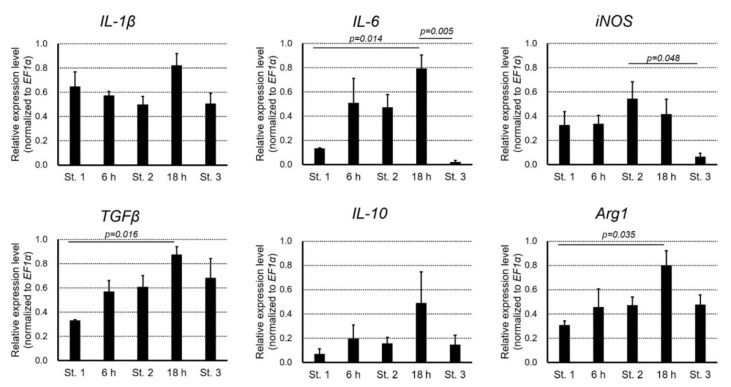
Changes in the relative expression levels of cytokines during re-epithelialization. Semi-quantitative PCR analysis was carried out with samples of the wound at Stage 1 (immediately after operation), 6 h, Stage 2 (12 h), 18 h, Stage 3 (96 h). The samples contained tissues in a range between the surface (containing the wound epidermis, if present) and a few millimeters depth within the wound area. IL-1β: interleukin-1 beta; IL-6: interleukin 6; iNOS: inducible nitric oxide synthase; TGFβ: transforming growth factor beta; IL-10: interleukin 10; Arg1: arginase 1; EF1α: elongation factor 1 alpha. The statistical significance among stages (*n* = 3 each) was analyzed by Tukey’s HSD multiple comparison test, and *p*-values less than 0.05 are illustrated. Sample images of electrophoresis of PCR products are shown in Appendix A.

**Figure 15 biomedicines-09-01892-f015:**
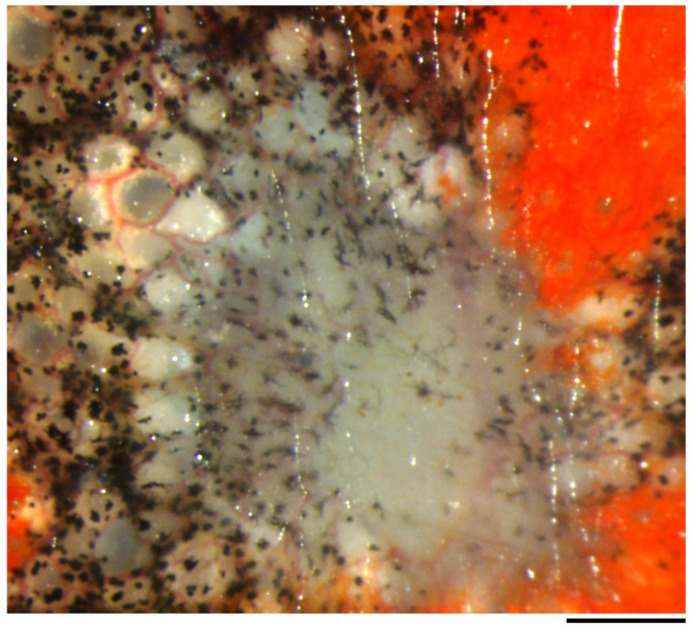
A representative image showing extension of blood capillaries from the wound margin. This image was taken at 28 days after full-thickness skin was excised from the patterned area on abdominal skin (see Appendix A). Blood capillaries for cutaneous respiration, which surrounded exocrine glands, extended from the wound margin, accompanying the migration of melanophores. Interestingly, capillaries were located under the region where single melanophores covered them. Scale bars: 500 μm.

**Figure 16 biomedicines-09-01892-f016:**
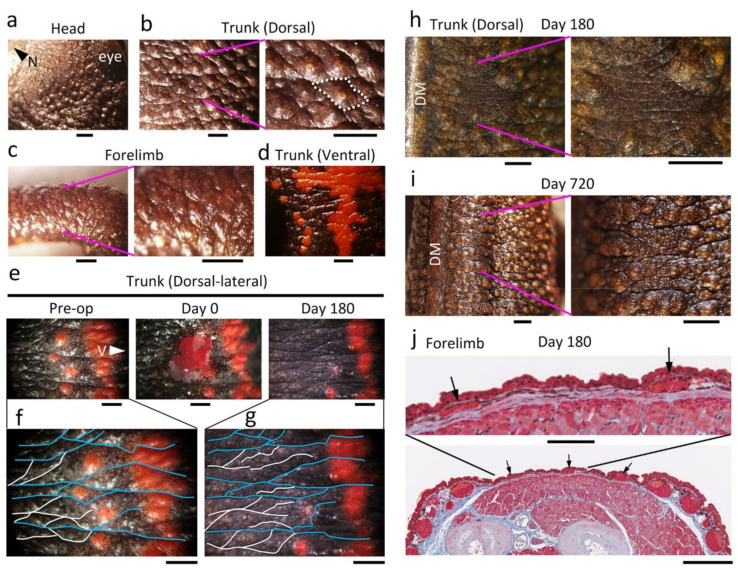
Restoration of skin texture and appendages. (**a**–**d**) Skin texture of normal skin. The surface of *C. pyrrhogaster* skin had structures such as the crista cutis and sulcus cutis. (**a**) Head. N: nasal side. (**b**) Dorsal skin of the trunk. One crista cutis surrounded by sulcus cutis (dotted line) is shown. An opening of the exocrine gland can be recognized on the top of the crista cutis. (**c**) Forelimb. (**d**) Ventral skin of the trunk. (**e**–**g**) A representative data set showing restoration of grooves on the dorsal skin of the trunk (*n* = 9). The dorsal skin near the border with the ventral skin (i.e., the dorsal-lateral skin) was excised. V: ventral side. (**f**,**g**) Enlargement of images before operation (Pre-op) and at 180 days after operation (Day 180). Thick and thin grooves were traced as blue and white lines, respectively. Note that the color pattern (orange spots) on lateral skin did not recover. (**h**) Representative data showing the appearance of the sulcus cutis on the wound (*n* = 9). Figure 1h was reproduced here. DM: dorsal midline. (**i**) Representative data showing recovery of skin texture (*n* = 2). We operated on adult newts as in (**h**) and selected those that exhibited incomplete recovery of skin texture at 180 days. Their skin wound was traced for as long as two years. In this animal, the wounded area near the DM blended into its surroundings by 720 days. Thick grooves had become restored across the wounded area. (**j**) A representative histological section of the wound in which the sulcus cutis had just appeared. Masson’s trichrome stain. At this stage, the layer of collagen-rich extracellular matrix (or reconstructing dermal layer) along the wound bed was still thinner than the dermis relative to the surroundings. Immature exocrine glands (arrows) were recognized under the presumptive crista cutis, and the epidermal region was flanked by thinner regions corresponding to the sulcus cutis. The characteristics of this tissue corresponded to Stage 6. Scale bars: 1 mm (**a**–**i**); 100 μm (upper panel in (**j**)); 400 μm (lower panel in (**j**)).

**Figure 17 biomedicines-09-01892-f017:**
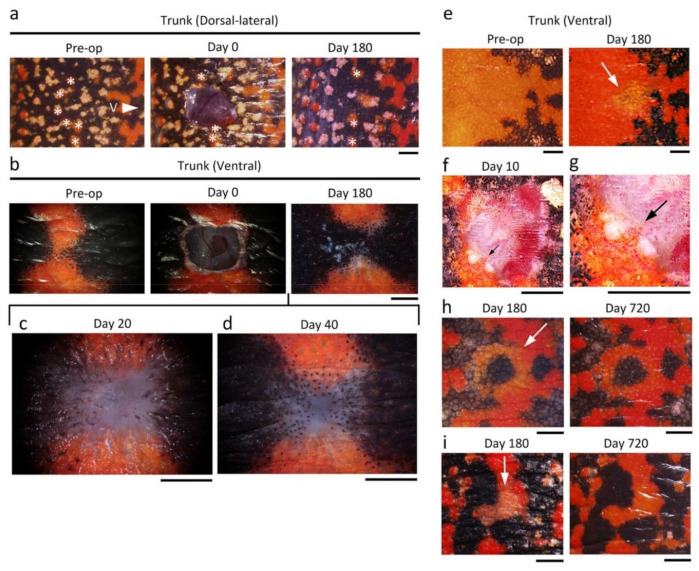
Restoration of skin color. (**a**) Sample image of dorsal-lateral skin (patterned area) of the trunk. This area contained a line of small orange spots (asterisks). The skin structure was almost restored by 180 days after operation (Day 180), whereas the orange spots on the excised skin were never restored; instead, the black area expanded into the wound region. V: ventral side. (**b**–**d**) Sample images of ventral skin (patterned area) of the trunk. A part of the ventral skin was excised across the black area (Day 0). Melanophores started collective migration from the wound margin (Day 20). This phase corresponds to Stage 5 in which collagen-rich extracellular matrix (reconstructing dermis) has filled the space between the wound epidermis and the wound bed. Collective migration of xanthophores started on Day 40. Melanophores detached from each other by this stage. Skin structure was almost restored by Day 180 whereas melanophores occupied most of the area of the wound region, resulting in an alteration of color pattern. (**e**) Sample image of ventral skin (plain orange area) of the trunk. A part of the orange area was excised without wounding the black area. Even though the structure of the skin had almost recovered by Day 180, the wound had not blended into its surroundings in terms of color tone (white arrow). (**f**,**g**) Sample image showing the collective migration of xanthophores at Day 10. Black arrows: a top of the extending orange area. (**h**,**i**) Sample images showing the recovery of the orange color tone in ventral skin. The color tone was monitored for longer than 2 years. The data in (**h**) was obtained from the animal shown in Figure 1i,j. In these two cases, the orange area created by Day 180 still had a pale tone (white arrow), but by Day 720 it became sufficiently dark to blend into its surroundings. Scale bars: 1 mm.

## Data Availability

All data used in this study are available from the corresponding authors upon reasonable request.

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
