# Peer review of "Skin Wound Healing of the Adult Newt, Cynops pyrrhogaster: A Unique Re-Epithelialization and Scarless Model"

_biomedicines, 2021, doi:10.3390/biomedicines9121892_

Round 1

Reviewer 1 Report

Dear Authors, 

The manuscript entitled "Skin Wound Healing of the Adult Newt, Cynops pyrrhogaster: A Unique Re-epithelialization and Scarless Model" addresses the study of skin regeneration without scar formation in wound healing. The authors have identified the stages of wound healing and skin regeneration on different body locations on adult newts. They concluded that the adult newts were able to regenerate the skin in a successful way in most of the anatomical zones examined. In contrast to humans, a granulation process during the re-epithelization of the skin was not observed in newts. This missing process is thought to be the reason for scarless skin regeneration. 

This interesting study sets the basis for regenerative medicine in human beings. The research is a very complete work gathering different techniques that are complementary to understand the healing process. 

The manuscript is well-written and well-organized, the literature included is of relevance. The methods are accurately described and the results obtained have been presented in a clear manner. 

I encourage the authors to conduct future studies on molecular and cellular levels to supplement the current research to progress to dermal reconstruction without scar formation. 

Author Response

To Reviewer #1:

Thank you for your encouraging comments. We will continue to investigate at the molecular and cellular levels the individual issues brought to light by the current study.

Reviewer 2 Report

In this manuscript titled “Skin Wound Healing of the Adult Newt, Cynops pyrrhogaster: A Unique Re-epithelialization and Scarless Model”, Ishii et al characterized a new wound healing model employing skin removal on adult Cynops pyrrhogaster. The authors first characterized the skin structure of the captured wild Cynops pyrrhogaster by histology, then surgery was performed to remove the full-thickness skin at several points of the body. Wound healing and re-epithelialization process was observed for more than two years. Dermal fibrosis, recovery of skin texture, appendages, and color, cell division, immune responses were evaluated. Through these experiments, the authors concluded that adult Cynops pyrrhogaster  could effectively regenerate their skin after a long recovery time. Overall, this is a well designed and nicely performed research. The authors made a thorough examination of the wound healing process in a species which was not examined in detail before. I only have several minor comments, which may further improve the quality of this study.

Minor issues:

  1. The timeline of re-epithelialization process is not well presented. The authors mostly showed representative pictures or histology graphs and showed a single time point counting results (Day 180, figure 3). From the scheme of figure 5, it is clear that the time needed for the full re-epithelization varies a lot among individual newts, ranging from 3 months to 6 months. Therefore, only showing representative pictures can not fully capture the variance of re-epithelialization time among the tested newt population. It would be nice if the authors can show how fast each individual newt achieved full re-epithelialization in a figure or table.

  1. Based on the text “We carefully selected one particular individual (CAGGs>mCherry (I-SceI) [12]) in which a cobble stone-shaped epithelial cell sheet (i.e., the stratum corneum) was observed on the surface of spots by its mCherry fluorescence (Figure 10a). This indicated that basal stem cells in the epidermis of the spot had been recombined, enabling us to monitor the movement of the epidermis along the wound edge during extension of the wound epidermis”, it seems that figure 10 is just based on a single newt. Drawing a conclusion from a single data point without experimental repeats is not scientifically sound and it will mislead readers. However, figure 10 is indeed an interesting experiment, the authors probably can move it into supplemental figure. 

  1. Please spend some time revising and improving the English writing of this manuscript. Some inaccurate English language here negatively affects the scientific rigor of this manuscript. For example, the authors used the term “metamorphosis” several times in the abstract and introduction to generally describe the development process of vertebrate. As far as I know, metamorphosis is a very specific term in development biology and amphibians are the only vertebrates which have metamorphosis process. It may be inaccurate to use this term on other vertebrates. Please double check and modify the language accordingly. As another example of inaccurate language, in the table of figure 3, the authors probably want to change “Texture”, “Appendages”, “Color” into “Texture Recovery”, “Appendage Recovery”, “Color Recovery”.

Author Response

To Reviewer #2:

1. The timeline of re-epithelialization process is not well presented. The authors mostly showed representative pictures or histology graphs and showed a single time point counting results (Day 180, figure 3). From the scheme of figure 5, it is clear that the time needed for the full re-epithelization varies a lot among individual newts, ranging from 3 months to 6 months. Therefore, only showing representative pictures cannot fully capture the variance of re-epithelialization time among the tested newt population. It would be nice if the authors can show how fast each individual newt achieved full re-epithelialization in a figure or table.

Response to 1:

The explanation of re-epithelialization in Figure 5 was probably a source of confusion. Therefore, we have revised the text (page 13, lines 1-4) as follows:

When the full-thickness skin was excised (Stage 1), re-epithelialization started with a multilayered epidermis (referred to as ‘wound epidermis’ here) which extended from the wound margin (Stage 2; 6-12 hours after operation), and completed as the wound epidermis covered the wound bed within 1-3 days (Stage 3). Following re-epithelialization, …

2. Based on the text “We carefully selected one particular individual (CAGGs>mCherry (I-SceI) [12]) in which a cobble stone-shaped epithelial cell sheet (i.e., the stratum corneum) was observed on the surface of spots by its mCherry fluorescence (Figure 10a). This indicated that basal stem cells in the epidermis of the spot had been recombined, enabling us to monitor the movement of the epidermis along the wound edge during extension of the wound epidermis”, it seems that figure 10 is just based on a single newt. Drawing a conclusion from a single data point without experimental repeats is not scientifically sound and it will mislead readers. However, figure 10 is indeed an interesting experiment, the authors probably can move it into supplemental figure.

Response to 2:

All the authors respectfully agree with Reviewer 2. However, this experiment was done to confirm the hypothesis derived from Figures 7 and 9, and as Reviewer 2 says, the results should not be overlooked even if they were obtained from a single newt. We are afraid that moving this to Supplementary Data will rather raise unnecessary suspicion. We believe that what we should do is to make it clear that this experiment needs to be repeated with multiple individuals to confirm the results.

We have added the following sentence on page 20, line 30:

To confirm the results of skin tracking, it is necessary to increase the number of available individuals in the future.

Furthermore, the following sentence was added to the Legend in Figure 10:

Note that this experiment was conducted on a single newt, so the same experiment needs to be repeated on multiple individuals to corroborate the results.

3-1. Please spend some time revising and improving the English writing of this manuscript. Some inaccurate English language here negatively affects the scientific rigor of this manuscript.

Response to 3-1:

We've tried our best. The manuscript has been edited by native speakers (SciRevision (http://www.scirevision.com/)) who have experience in reviewing many scientific papers.

3-2. For example, the authors used the term “metamorphosis” several times in the abstract and introduction to generally describe the development process of vertebrate. As far as I know, metamorphosis is a very specific term in development biology and amphibians are the only vertebrates which have metamorphosis process. It may be inaccurate to use this term on other vertebrates. Please double check and modify the language accordingly.

Response to 3-2:

We have revised the explanation of life stages of tetrapods (page 2, lines 39-42) as follows:

As a rule, four-limbed vertebrates (tetrapods), including humans, have a relatively high capacity to repair or regenerate their injured body parts in their aquatic life-stages (as larvae (amphibians), embryos (reptiles and birds) or fetuses (mammals)), but this ability declines or is lost as they transit to terrestrial life-stages through metamorphosis (amphibians), hatching (reptiles and birds) or birth (mammals) [4-12].

3-3. As another example of inaccurate language, in the table of figure 3, the authors probably want to change “Texture”, “Appendages”, “Color” into “Texture Recovery”, “Appendage Recovery”, “Color Recovery”.

Response to 3-3:

We have corrected the table of Figure 3b according to the comment.